# 3D Siamese Voxel-to-BEV Tracker for Sparse Point Clouds

**Le Hui**[†], **Lingpeng Wang**[†], **Mingmei Cheng, Jin Xie**[∗], **Jian Yang**[∗]
PCA Lab, Nanjing University of Science and Technology, China
{le.hui, cslpwang, chengmm, csjxie, csjyang}@njust.edu.cn

## Abstract

3D object tracking in point clouds is still a challenging problem due to the sparsity of LiDAR points in dynamic environments. In this work, we propose a Siamese voxel-to-BEV tracker, which can significantly improve the tracking performance in sparse 3D point clouds. Specifically, it consists of a Siamese shape-aware feature learning network and a voxel-to-BEV target localization network. The Siamese shape-aware feature learning network can capture 3D shape information of the object to learn the discriminative features of the object so that the potential target from the background in sparse point clouds can be identified. To this end, we first perform template feature embedding to embed the template's feature into the potential target and then generate a dense 3D shape to characterize the shape information of the potential target. For localizing the tracked target, the voxel-to-BEV target localization network regresses the target's 2D center and the $z$-axis center from the dense bird's eye view (BEV) feature map in an anchor-free manner. Concretely, we compress the voxelized point cloud along $z$-axis through max pooling to obtain a dense BEV feature map, where the regression of the 2D center and the $z$-axis center can be performed more effectively. Extensive evaluation on the KITTI and nuScenes datasets shows that our method significantly outperforms the current state-of-the-art methods by a large margin. Code is available at https://github.com/fpthink/V2B.

## 1 Introduction

Object tracking is an essential task in computer vision and has been widely in various applications, such as autonomous vehicle, mobile robotics, and augmented reality. In the past few years, many efforts [33, 2, 12, 64] have been made on 2D object tracking from RGB data. Recently, with the development of 3D sensor such as LiDAR and Kinect, 3D object tracking [37, 72, 53, 28, 42] has attracted more attention. Lately, some pioneering works [22, 18, 52] have focused on point cloud based 3D object tracking. However, due to the sparsity of 3D point clouds, 3D object tracking on point clouds is still a challenging task.

Few works are dedicated to 3D single object tracking (SOT) with only point clouds. As a pioneer, SC3D [21] is the first 3D Siamese tracker that performs matching between the template and candidate 3D target proposals generated by Kalman filtering [22]. Furthermore, a shape completion network is used to enhance shape information of candidate proposals in sparse point clouds, thereby improving the accuracy of matching. However, SC3D cannot perform the end-to-end training, and consumes

---

[†]Equal Contributions, [∗]Corresponding authors.
Le Hui, Lingpeng Wang, Mingmei Cheng, Jin Xie, and Jian Yang are with PCA Lab, Key Lab of Intelligent Perception and Systems for High-Dimensional Information of Ministry of Education, and Jiangsu Key Lab of Image and Video Understanding for Social Security, School of Computer Science and Engineering, Nanjing University of Science and Technology, China.

35th Conference on Neural Information Processing Systems (NeurIPS 2021).

much time when matching exhaustive candidate proposals. Towards these concerns, Qi *et al.* [52] proposed an end-to-end framework termed P2B, which first localizes potential target centers in the search area via Hough voting [48], and then aggregates vote clusters to generate target proposals. Nonetheless, when facing sparse scenes, P2B may not be able to track the object accurately, or even lose the tracked object. On the one hand, it adopts random sampling to generate initial seed points, which further exacerbates the sparsity of point clouds. On the other hand, it is difficult to generate high-quality target proposals on sparse 3D point clouds. Although SC3D has enhanced shape information of candidate proposals, the low-quality candidate proposals obtained from sparse point clouds still degrade tacking performance.

As shown in Fig. 1, we count the number of points on KITTI's cars. It can be found that 51% of cars have less than 100 points, and only 7% of cars have more than 2500 points. When facing sparse point clouds, it is difficult to distinguish the target from the background due to the sparsity of point clouds. Therefore, how to improve the tracking performance in sparse scenes should be considered. Our intuition consists of two folds. First, enhancing shape information of target will provide discriminative information to distinguish the target from the background, especially in sparse point clouds. Second, due to the sparsity of the point cloud, it is difficult to regress the target center in 3D space.

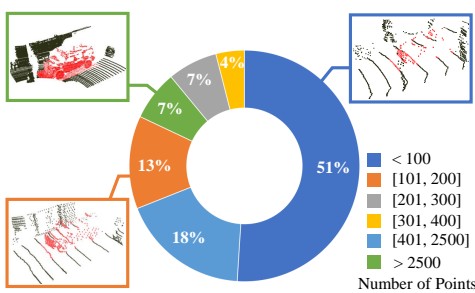

Figure 1: Statistics of the number of points on KITTI's cars. Cars are colored in red.

We hence consider compressing the sparse 3D space into a dense 2D space, and perform center regression in the dense 2D space to improve tracking performance.

In this paper, we propose a novel Siamese voxel-to-BEV (V2B) tracker, which aims to improve the tracking performance of 3D single object tracking, especially in sparse point clouds. We illustrate our framework in Fig. 2. We first feed the template and search area into the Siamese network to extract point features, respectively. Then, we employ the global and local template feature embedding to strengthen the correlation between the template and search area so that the potential target in the search area can be effectively localized. After that, we introduce a shape-aware feature learning module to learn the dense geometric features of the potential target, where the complete and dense point clouds of the target are generated. Thus, the geometric structures of the potential target can be captured better so that the potential target can be effectively distinguished from the background in the search area. Finally, we develop a voxel-to-BEV target localization network to localize the target in the search area. In order to avoid using the low-quality proposals on sparse point clouds for target center prediction, we directly regress the 3D center of the target with the highest response in the dense bird's eye view (BEV) feature map, where the dense BEV feature map is generated by voxelizing the learned dense geometric features and performing max-pooling along the $z$ axis. Thus, with the constructed dense BEV feature map, for sparse point clouds, our method can more accurately localize the target center without any proposal.

In summary, we propose a novel Siamese voxel-to-BEV tracker, which can significantly improve tracking performance, especially in sparse point clouds. We develop a Siamese shape-aware feature learning network that can introduce shape information to enhance the discrimination of the potential target in the search area. We develop a voxel-to-BEV target localization network, which can accurately detect the 3D target's center in the dense BEV space compared to sparse 3D space. Extensive results show that our method has achieved new state-of-the-art results on the KITTI dataset [20], and has a good generalization ability on the nuScenes [6] dataset.

## 2 Related Work

**2D object tracking.** Numerous schemes [5, 22, 34, 74, 1] have been presented and achieved impressive results in 2D object tracking. Early works are mainly based on correlation filtering. As a pioneer, MOSSE [4] presents stable correlation filters for visual tracking. After that, correlation-based methods use Circulant matrices [25], kernelized correlation filters [26], continuous convolution filters [13], factorized convolution operators [12] to improve tracking performance. In recent years, Siamese-based methods [16, 24] have been more popular in the tracking field. In [2], Bertinetto *et*

*al.* proposed SiamFC, a pioneering work that combines naive feature correlation with a fully-convolutional Siamese network for object tracking. Subsequently, some improvements [85, 68, 75, 82, 76] are made to Siamese trackers, such as combining with a region proposal network [17, 39, 80, 65] or an anchor-free FCOS detector [11], using a deeper architecture [38] or two-branch structure [23], exploiting attention [67, 84] or self-attention [7], applying triplet loss [14]. However, these methods are specially designed for 2D object tracking, so they cannot be directly applied to 3D point clouds.

**3D single object tracking.** Early 3D single object tracking (SOT) methods focus on RGB-D information. As a pioneer, Song *et al.* [59] first proposed a unified 100 RGB-D video dataset, which opened up a new research direction for RGB-D tracking [3, 43, 29]. Based on RGB-D information, 3D SOT methods [60, 45, 53] usually combines techniques from the 2D tracking with additional depth information. However, RGB-D tracking also relies on RGB information, and it may fail when the RGB information is degraded. Recent efforts [46, 69] begin to use LiDAR point clouds for 3D single object tracking. Among them, SC3D [21] is the first 3D Siamese tracker, but it is not an end-to-end framework. Following it, Re-Track [18] is a two-stage framework that re-tracks the lost objects of the coarse stage in the fine stage. Lately, Qi *et al.* [52] proposed P2B, which solves the problem that SC3D cannot perform end-to-end training and consumes a lot of time. P2B uses VoteNet [48] to generate proposals and selects the proposal with the highest score as the target. Based on P2B, to handle sparse and incomplete target shapes, BAT [83] introduces a box-aware feature module to enhance the correlation learning between template and search area. Nonetheless, when facing very sparse scenarios, VoteNet used in P2B and BAT may be difficult to generate high-quality proposals, resulting in performance degradation.

**3D multi-object tracking.** Most 3D multi-object tracking (MOT) systems follow the same schemes with the 2D multi-object tracking systems, but the only difference is that 2D detection methods are replaced by 3D detection methods. Most 3D MOT methods [73, 55, 30] usually adopt tracking-by-detection schemes. Specifically, they first use a 3D object detector [57, 58, 56] to detect numerous objects of each frame, and then exploit the data association between detection results of two frames to match the corresponding objects. To exploit the data association, early works [54] use handcrafted features such as spatial distance. Instead, modern 3D trackers use motion information that can be obtained by 3D Kalman filters [47, 10, 71] and learned deep features [78, 81].

**Deep learning on point clouds.** With the introduction of PointNet [49], 3D deep learning on point clouds has stimulated the interest of researchers. Existing methods can be mainly divided into: point-based [51, 40, 63, 9], volumetric-based [50, 44], graph-based [70, 36, 35, 66, 8, 27], and view-based [61, 62, 77] methods. However, volumetric-based and view-based methods lose fine-grained geometric information due to voxelization and projection, while graph-based methods are not suitable for sparse point clouds since few points cannot provide sufficient local geometric information for constructing a graph. Thus, existing 3D tracking networks [21, 18, 52, 83] are point-based methods.

## 3 Method

Our work is specifically designed for 3D single object tracking in sparse point clouds. An overview of our framework is depicted in Fig. 2. We first present the Siamese shape-aware feature learning network to enhance the discrimination of the potential target in search area (Sec. 3.1). We then localize the target by voxel-to-BEV target localization network (Sec. 3.2).

### 3.1 Siamese Shape-Aware Feature Learning Network

#### 3.1.1 Template Feature Embedding

Suppose the size of the template is $N$, and the size of the search area is $M$ (generally, $M > N$). Before template feature embedding, we first use the Siamese network to extract point features of the template and search area, denoted by $P = \{\boldsymbol{p}_i\}_{i=1}^{N}$ and $Q = \{\boldsymbol{q}_j\}_{j=1}^{M}$. The Siamese network consists of template branch and detection branch. In order to reduce the inference time of the network, we just use PointNet++ [51] as the backbone and share parameters. It can be replaced with a powerful network such as KPConv [63]. We then employ template feature embedding to encode the search area by learning the similarity of the global shape and local geometric structures between the template and search area. The illustration of template feature embedding is shown in the left half of Fig. 3.

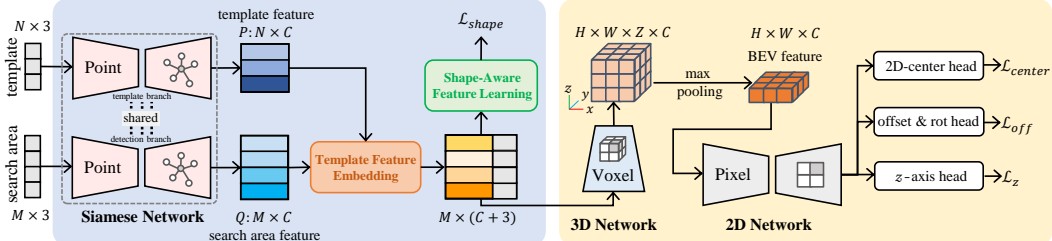

Figure 2: The architecture of V2B. Given a template and search area, we first use Siamese network to obtain template and search area features. We then perform template feature embedding and shape-aware feature learning to enhance the ability to distinguish the target from the background. Finally, we perform voxel-to-BEV target localization to detect the 3D object center from the BEV.

**Template global feature embedding.** We use the multi-layer perceptron (MLP) network to adaptively learn the correlation between the template and search area. The similarity between the template and search area is formulated as:

$$\boldsymbol{w}_{ij} = f_{corr}(\boldsymbol{p}_i, \boldsymbol{q}_j) = \text{MLP}(\boldsymbol{p}_i - \boldsymbol{q}_j), \forall \boldsymbol{p}_i \in P, \boldsymbol{q}_j \in Q \tag{1}$$

where $\boldsymbol{p}_i - \boldsymbol{q}_j$ characterizes the difference between the two feature vectors and $\boldsymbol{w}_{ij} \in \mathbb{R}^C$ is the correlation weight between two points. The global shape information of the template is given by:

$$\boldsymbol{q}_j' = f_{emb}(\boldsymbol{q}_j, \boldsymbol{p}_1, \boldsymbol{p}_2, \dots, \boldsymbol{p}_N) = \text{MLP}(\underset{i=1,2,\dots,N}{\text{MAX}}\{\boldsymbol{p}_i \cdot \boldsymbol{w}_{ij}\}), \forall \boldsymbol{q}_j \in Q \tag{2}$$

where MAX represents the max pooling function and $\boldsymbol{w}_{ij}$ is the correlation weight. The obtained $\boldsymbol{q}_j' \in \mathbb{R}^C$ considers the similarity between the template and search area, and characterizes the global shape information of the target through the max pooling function.

**Template local feature embedding.** To characterize the local similarity between the template and search area, we first obtain the similarity map by computing the cosine distance between them. The similarity function $f_{sim}$ is written as:

$$\boldsymbol{s}_{ij} = f_{sim}(\boldsymbol{p}_i, \boldsymbol{q_j}) = \frac{\boldsymbol{p}_i^\top \cdot \boldsymbol{q_j}}{\|\boldsymbol{p}_i\|_2 \cdot \|\boldsymbol{q}_j\|_2}, \forall \boldsymbol{p}_i \in P, \boldsymbol{q}_j \in Q \tag{3}$$

where $\boldsymbol{s}_{ij}$ indicates the similarity between points $i$ and $j$. We then assign each point in the search area with its most similar point in the template, which is written as:

$$\boldsymbol{q}_j'' = \text{MLP}([\boldsymbol{q}_j, \boldsymbol{s}_{kj}, \boldsymbol{p}_k, \boldsymbol{x}_k]), k = \underset{i=1,2,\dots,N}{\text{argmax}}\{f_{sim}(\boldsymbol{p}_i, \boldsymbol{q}_j)\}, \forall \boldsymbol{q}_j \in Q \tag{4}$$

where $k$ indicates the index of the maximum value of similarity, and $\boldsymbol{s}_{kj}$, $\boldsymbol{x}_k$ are the corresponding maximum value and 3D coordinate, respectively. $[\cdot, \cdot, \cdot, \cdot]$ represents the concatenation operator. We hence obtain the embedded feature $\boldsymbol{q}_j'' \in \mathbb{R}^C$ of $j$-th point in the search area after using MLP. Finally, we concatenate the obtained global and local feature maps to obtain an enhanced feature map $F = \{\boldsymbol{f}_j\}_{j=1}^M, \boldsymbol{f}_j = \text{MLP}[\boldsymbol{q}_j', \boldsymbol{q}_j'']$.

### 3.1.2 Shape-Aware Feature Learning

Due to the sparse and incomplete point clouds of the potential target in the search area, we employ shape-aware feature learning to learn dense geometric features of the target, where the dense and complete point clouds of the target can be obtained. It is expected that the learned features from the generated dense point clouds can characterize the geometric structures of the target better.

**Dense ground truth processing.** To obtain the dense 3D point cloud ground truth, we first crop and center points lying inside the target's ground truth bounding box in all frames. We then concatenate all cropped and centered points to generate a dense aligned 3D point cloud, denoted by $X = \{\boldsymbol{x}_i\}_{i=1}^{2048}$, where $\boldsymbol{x}_i$ is the 3D position, and we fix the number of points to 2048 by randomly discarding and duplicating points.

**Shape information encoding.** We depict the network structure in the right half of Fig. 3. Suppose the input point feature $\boldsymbol{F} \in \mathbb{R}^{M \times C}$ that has been embedded with the template information. Before generating a dense and complete point cloud of the target, we first use a gate mechanism to enhance

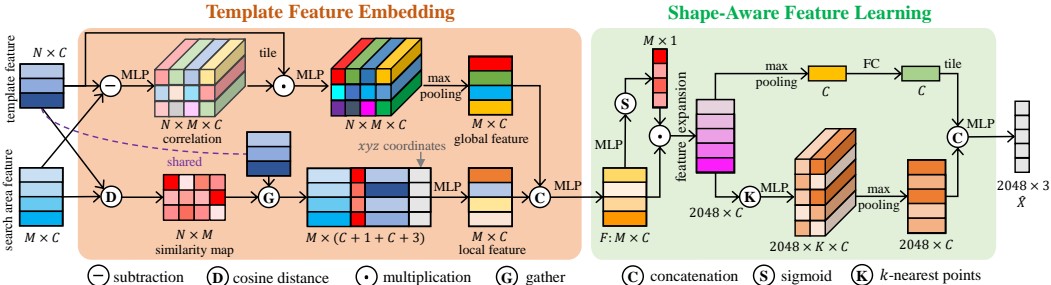

Figure 3: The architecture of template feature embedding and shape-aware feature learning.

the feature of the potential target and suppress the background in the search area, which is written as:

$$\boldsymbol{F}^{'} = \sigma(\boldsymbol{F}\boldsymbol{W}^{\top} + b) \circ \boldsymbol{F} \tag{5}$$

where $\boldsymbol{F}^{'} \in \mathbb{R}^{M \times C}$ is the enhanced feature map, $\sigma$ is the *sigmoid* function, and $\circ$ is the element-wise product. Besides, $\boldsymbol{W} \in \mathbb{R}^{1 \times C}$ is the weight to be learned. It is expected that the potential target can provide more information for generating a dense and complete point cloud of the target. Once we obtain the enhanced feature map, we execute the feature expansion operation [79] to enlarge the feature map from $M \times C$ to $2048 \times C$, *i.e.*, the number of points increases from $M$ to $2048$. Then, we capture the global shape information and local geometric structure of the potential target to generate the complete and dense 3D shape. On the one hand, we exploit the max pooling combined with fully connected layers to capture global shape information. On the other hand, we adopt EdgeConv [70] to capture local geometric information of the target. After that, we augment the local feature of each point with the global shape information, yielding a new feature map of size $2048 \times 2C$. Finally, we adopt MLP to generate 3D coordinates, denoted by $\hat{X} = \{\hat{\boldsymbol{x}}_i\}_{i=1}^{2048}$. To train the shape generation network, we follow [21, 15] and use Chamfer distance (CD) loss to enforce the network to generate a realistic 3D point cloud. The Chamfer distance measures the similarity between the generated point cloud and dense ground truth, which is given by:

$$\mathcal{L}_{shape} = \sum_{\boldsymbol{x}_i \in X} \min_{\hat{\boldsymbol{x}}_j \in \hat{X}} \|\boldsymbol{x}_i - \hat{\boldsymbol{x}}_j\|_2^2 + \sum_{\hat{\boldsymbol{x}}_j \in \hat{X}} \min_{\boldsymbol{x}_i \in X} \|\boldsymbol{x}_i - \hat{\boldsymbol{x}}_j\|_2^2 \tag{6}$$

By minimizing the CD loss, we can learn dense geometric features of the potential target in the search area by generating a dense and complete point cloud of the target. Note that shape information encoding is only performed during training and will be discarded during testing. Thus, it does not increase the inference time of object tracking in the test scheme.

Although SC3D [21] also uses shape completion to encode shape information of the target, the template completion model in SC3D cannot recover complex geometric structures of the potential target well due to limited templates and large variations of the potential target in the search area. However, our method is a complex point cloud generation method that learns the target completion model from the samples of search areas with the gate mechanism to enhance the feature of the potential target and suppress the background in the search area. In addition, the template completion model in SC3D only employs PointNet [49] to extract point features of sparse point clouds, while our target completion model constructs a global-local branch to extract global shape features and local geometric features of sparse point clouds.

## 3.2 Voxel-to-BEV Target Localization Network

In order to avoid using the low-quality proposals on sparse point clouds for target center prediction, we develop a simple yet effective target center localization network without any proposal to improve the localization precision in sparse point clouds.

### 3.2.1 Dense BEV Feature Map Generation

In order to improve the localization precision in sparse point clouds, we utilize the voxelization and max-pooling operation to convert the learned discriminative features of sparse 3D points into the dense bird's eye view (BEV) feature map for the target localization, as shown in the right half of Fig. 2. We first convert the point features of the search area into a volumetric representation by

averaging the 3D coordinates and features of the points in the same voxel bin. Then, we apply a stack of 3D convolutions on the voxelized feature map to aggregate the feature of the potential target in the search area, where the voxels lying on the target can be encoded with rich target information. However, in the sparse volume space, due to the large number of empty voxels, the differences between the responses in the voxelized feature map might not be remarkable. Thus, the highest response in the feature map is difficult to distinguish from the low responses, leading to the inaccurate regression of the 3D center of the target, including the $z$-axis center. By performing max-pooling on the voxelized feature map along the $z$-axis, we can obtain the dense BEV feature map, where the low responses in the voxelized feature map can be suppressed. Thus, compared to the voxelized feature map, we can more accurately localize the 2D center of the target with the highest response in the dense BEV feature map. The response of the 2D center (*i.e.*, max-pooling feature along the $z$-axis) in the BEV feature map actually contains the geometric structure information of the potential target while the responses of other points in the BEV feature map do not. In addition, we apply a stack of 2D convolutions on the dense BEV feature map to aggregate the feature so that the potential target can obtain sufficient local information in the BEV feature map. Thus, with the constructed dense BEV feature map, for sparse point clouds, our method can more accurately localize the target center without any proposal.

### 3.2.2 Target Localization in BEV

Inspired by [19], we develop a simple yet powerful network to detect the 2D center and the $z$-axis center based on the obtained dense BEV feature map. As shown in the right half of Fig. 2, it consists of three heads: 2D-center head, offset & rotation head, and $z$-axis head. The 2D-center head aims to localize 2D center of target on the $x$-$y$ plane, and $z$-axis head regresses the target center of the $z$-axis. Since the 2D center of 2D grid is discrete, we also regress the offset between it and the continuous center. Thus, we use an offset & rotation head to regress offset plus additional rotation.

**Target center parameterization.** Given the voxel size $v$ and the range of the search area $[(x_{min}, x_{max}), (y_{min}, y_{max})]$ in $x$-$y$ plane, we can obtain the resolution of the BEV feature map by $H = \lfloor \frac{x_{max}-x_{min}}{v} \rfloor + 1$ and $W = \lfloor \frac{y_{max}-y_{min}}{v} \rfloor + 1$, where $\lfloor \cdot \rfloor$ is the floor operation. Assuming the 3D center $(x, y, z)$ of the target ground truth, we can compute the 2D target center $c = (c_x, c_y)$ in $x$-$y$ plane by $c_x = \frac{x-x_{min}}{v}$ and $c_y = \frac{y-y_{min}}{v}$. Besides, the discrete 2D center $\tilde{c} = (\tilde{c}_x, \tilde{c}_y)$ is defined by $\tilde{c}_x = \lfloor c_x \rfloor$ and $\tilde{c}_y = \lfloor c_y \rfloor$.

**2D-center head.** Following [19], we obtain the target center's ground truth $\mathcal{H} \in \mathbb{R}^{H \times W \times 1}$. For the pixel $(i, j)$ in the 2D bounding box, if $i = \tilde{c}_x$ and $j = \tilde{c}_y$, the $\mathcal{H}_{ij} = 1$, otherwise $\frac{1}{d+1}$, where $d$ represents the Euclidean distance between the pixel $(i, j)$ and the target center $(\tilde{c}_x, \tilde{c}_y)$. For any pixel outside the 2D bounding box, $\mathcal{H}_{ij}$ is set to 0. In the training phase, we enforce the predicted map $\hat{\mathcal{H}} \in \mathbb{R}^{H \times W \times 1}$ to approach the ground truth $\mathcal{H}$ by using Focal loss [41]. The modified Focal loss is formulated as:

$$\mathcal{L}_{center} = -\sum \mathbb{I}[\mathcal{H}_{ij} = 1] \cdot (1-\hat{\mathcal{H}}_{ij})^\alpha \log(\hat{\mathcal{H}}_{ij}) + \mathbb{I}[\mathcal{H}_{ij} \neq 1] \cdot (1-\mathcal{H}_{ij})^\beta (\hat{\mathcal{H}}_{ij})^\alpha \log(1-\hat{\mathcal{H}}_{ij}) \quad (7)$$

where $\mathbb{I}(cond.)$ is the indicator function. If *cond.* is true, then $\mathbb{I}(cond.) = 1$, otherwise 0. Besides, we empirically set $\alpha = 2$ and $\beta = 4$ in all experiments.

**Offset & rotation head.** Since the continuous 2D object center is converted into the discrete one by floor operation, we consider regressing the offset of the continuous ground truth center. To improve the accuracy of regression, we consider a square area with radius $r$ around the object center. Here, we also add rotation regression. Given a predicted map $\hat{\mathcal{O}} \in \mathbb{R}^{H \times W \times 3}$, where 3-dim means the 2D coordinate offset plus rotation, the error of offset and rotation is expressed as:

$$\mathcal{L}_{off} = \sum_{\triangle x=-r}^{r} \sum_{\triangle y=-r}^{r} \left| \hat{\mathcal{O}}_{\tilde{c}+(\triangle x, \triangle y)} - [c - \tilde{c} + (\triangle x, \triangle y), \theta] \right| \quad (8)$$

where $\tilde{c}$ and $c$ mean the discrete and continuous position of the ground truth center, respectively. Besides, $\theta$ indicates the ground truth rotation angle and $[\cdot, \cdot]$ is the concatenation operation.

$z$**-axis head.** We directly regress the $z$-axis location of the target center from the BEV feature map. Given a predicted map $\hat{\mathcal{Z}} \in \mathbb{R}^{H \times W \times 1}$, we use $L_1$ loss to compute the error of $z$-axis center by:

$$\mathcal{L}_z = \left| \hat{\mathcal{Z}}_{\tilde{c}} - z \right| \quad (9)$$

where $\tilde{c}$ is the discrete object center, and $z$ is $z$-axis center's ground truth.

The final loss of our network is as follows: $\mathcal{L}_{total} = \lambda_1\mathcal{L}_{shape} + \lambda_2(\mathcal{L}_{center} + \mathcal{L}_{off}) + \lambda_3\mathcal{L}_z$, where $\lambda_1$, $\lambda_2$, and $\lambda_3$ are the hyperparameter for shape generation, 2D center and offset regression, and $z$-axis position regression, respectively. In the experiment, we set $\lambda_1 = 10^{-6}$, $\lambda_2 = 1.0$, and $\lambda_3 = 2.0$.

## 4 Experiments

### 4.1 Experimental Settings

**Datasets.** For 3D single object tracking, we use KITTI [20] and nuScenes [6] datasets for training and evaluation. Since the ground truth of the test set of KITTI dataset cannot be obtained, we follow [21, 52] and use the training set to train and evaluate our method. It contains 21 video sequences and 8 types of objects. We use scenes 0-16 for training, scenes 17-18 for validation, and scenes 19-20 for testing. For nuScenes dataset, we use its validation set to evaluate the generalization ability of our method. Note that the nuScenes dataset only labels key frames, so we report the performance evaluated on the key frames.

**Evaluation metrics.** For 3D single object tracking, we use the *Success* and *Precision* defined in the one pass evaluation (OPE) [32] to evaluate the tracking performance of different methods. *Success* measures the IOU between the predicted and ground truth bounding boxes, while *Precision* measures the error AUC of the distance between the centers of two bounding boxes.

**Implementation details.** Following [52], we set the number of points $N = 512$ and $M = 1024$ for the template and search area by randomly discarding and duplicating points. For the backbone network, we use a slightly modified PointNet++ [51], which consists of three set-abstraction (SA) layers (with query radius of 0.3, 0.5, and 0.7) and three feature propagation (FP) layers. For each SA layer passed, the points will be randomly downsampled by half. For the shape generation network, we generate 2048 points. The global branch is the max pooling combined with two fully connected layers, while the local branch only uses one EdgeConv layer. We use a two layer MLP network to generate 3D coordinates. For 3D center detection, the voxel size is set to 0.3 meters in volumetric space. We stack four 3D convolutions (with stride of 2, 1, 2, 1 along the $z$-axis) and four 2D convolutions (with stride of 2, 1, 1, 2) combined with the skip connections for feature aggregation, respectively. For all experiments, we use Adam [31] optimizer with learning rate 0.001 for training, and the learning rate decays by 0.2 every 6 epochs. It takes about 20 epochs to train our model to convergence.

**Training and testing.** For training, we combine the points inside the first ground truth bounding box (GTBB) and the points inside the previous GTBB plus the random offset as the template of the current frame. To generate the search area, we enlarge the current GTBB by 2 meters and plus the random offset. For testing, we fuse the points inside the first GTBB and the previous result's point cloud (if exists) as the template. Besides, we first enlarge the previous result by 2 meters in current frame, and then collect the points lying in it to generate the search area.

### 4.2 Results

**Quantitative results.** We compare our method with current state-of-the-art methods, including SC3D [21], P2B [52], and BAT [83]. The quantitative results are listed in Tab. 1. For the KITTI [20] dataset, we follow [21, 52] and report the performance of four categories, including car, pedestrian, van, and cyclist, and their average results. As one can see from the table, our method is significantly better than other methods on the mean results of four categories. For the car category, our method can even improve the *Success* from 60.5% (BAT) to 70.5% (V2B). For the nuScenes [6] dataset, we directly apply the models, trained on the corresponding categories of the KITTI dataset, to evaluate performance on the nuScenes dataset. Specifically, the corresponding categories between KITTI and nuScenes datasets are Car→Car, Pedestrian→Pedestrian, Van→Truck, and Cyclist→Bicycle, respectively. It can be seen that our V2B can still achieve better performance on the mean results of all four categories. The quantitative results on the nuScenes dataset further demonstrate that our V2B has a good generalization ability to adapt to different datasets.

**Quantitative results on sparse scenes.** To verify the effectiveness of our method for object tracking in sparse scenes, we count the performance of SC3D, P2B, BAT, and our V2B in sparse scenes of the

Table 1: The *Success*/*Precision* of different methods on the KITTI and nuScenes datasets. "Mean" indicates the average results of four categories.

| Dataset | Method
Frame Number | Car
6424 | Pedestrian
6088 | Van
1248 | Cyclist
308 | Mean
14068 |
|---|---|---|---|---|---|
| KITTI | SC3D [21] | 41.3 / 57.9 | 18.2 / 37.8 | 40.4 / 47.0 | **41.5 / 70.4** | 31.2 / 48.5 |
|  | P2B [52] | 56.2 / 72.8 | 28.7 / 49.6 | 40.8 / 48.4 | 32.1 / 44.7 | 42.4 / 60.0 |
|  | BAT [83] | 60.5 / 77.7 | 42.1 / 70.1 | **52.4 / 67.0** | 33.7 / 45.4 | 51.2 / 72.8 |
|  | V2B (ours) | **70.5 / 81.3** | **48.3 / 73.5** | 50.1 / 58.0 | 40.8 / 49.7 | **58.4 / 75.2** |

| Dataset | Method
Frame Number | Car
15578 | Pedestrian
8019 | Truck
3710 | Bicycle
501 | Mean
27808 |
|---|---|---|---|---|---|
| nuScenes | SC3D [21] | 25.0 / 27.1 | 14.2 / 16.2 | **25.7 / 21.9** | 17.0 / 18.2 | 21.8 / 23.1 |
|  | P2B [52] | 27.0 / 29.2 | 15.9 / 22.0 | 21.5 / 16.2 | 20.0 / **26.4** | 22.9 / 25.3 |
|  | BAT [83] | 22.5 / 24.1 | **17.3 / 24.5** | 19.3 / 15.8 | 17.0 / 18.8 | 20.5 / 23.0 |
|  | V2B (ours) | **31.3 / 35.1** | **17.3** / 23.4 | 21.7 / 16.7 | **22.2** / 19.1 | **25.8 / 29.0** |

KITTI dataset. Specifically, we filter out sparse scenes for evaluation according to the number of points lying in the target bounding boxes in the test set. Specifically, the conditions for the sparse scenes are: $\leq 150$ (car), $\leq 100$ (pedestrian), $\leq 150$ (van), and $\leq 100$ (cyclist), respectively. For the four categories, the number of selected frames are 3293 (car), 1654 (pedestrian), 734 (van), and 59 (cyclist), respectively. In Tab. 2, we report the results of *Success* and *Precision*. As one can see from the table, our V2B achieves the best performance on the mean results of all four categories. Note that when switching from sparse frames (Tab. 2) to all types of frames (Tab. 1), SC3D and P2B suffer from a performance drop on the mean results of four categories. The worse tracking performance of SC3D and P2B on large amounts of sparse frames leads to the inaccurate template updates on the consecutive dense frames. Thus, SC3D and P2B cannot obtain better tracking performance on the dense frames. Although SC3D uses template shape completion, due to limited template samples and large variations of the potential target in the search area, it cannot accurately recover the complex geometric structures of the target in the sparse frames, which poses challenges on localizing the potential target with sparse points. On the contrary, our V2B employs the proposed shape-aware feature learning module to generate dense and complete point clouds of the potential target for the target shape completion, leading to more accurate localization of the target in the sparse frames. Compared with BAT, our V2B achieves the performance gain of 2% on the mean results of all four categories from sparse frames to all types of frames. Therefore, the comparison results can demonstrate that our V2B can effectively improve the performance of single object tracking in sparse point clouds.

Table 2: Comparison of *Success*/*Precision* of different methods on the **sparse** scenarios.

| Method
Frame Number | Car
3293 | Pedestrian
1654 | Van
734 | Cyclist
59 | Mean
5740 |
|---|---|---|---|---|---|
| SC3D [21] | 37.9 / 53.0 | 20.1 / 42.0 | 36.2 / 48.7 | **50.2 / 69.2** | 32.7 / 49.4 |
| P2B [52] | 56.0 / 70.6 | 33.1 / 58.2 | 41.1 / 46.3 | 24.1 / 28.3 | 47.2 / 63.5 |
| BAT [83] | 60.7 / 75.5 | 48.3 / **77.1** | 41.5 / 47.4 | 25.3 / 30.5 | 54.3 / 71.9 |
| V2B (ours) | **64.7 / 77.4** | **50.8** / 74.2 | **46.8 / 55.1** | 30.4 / 37.2 | **58.0 / 73.2** |

**Visualization results.** As shown in Fig. 4, we plot the visualization results of P2B and our V2B on the car category. Specifically, we plot a couple sparse and dense scenarios on the car category of the KITTI dataset. It can be clearly seen from the figure that compared with P2B, our V2B can track the targets more accurately in both sparse and dense scenes. Especially in sparse scenes, compared with P2B, our V2B can track the targets effectively. The visualization results can demonstrate the effectiveness of our V2B for sparse point clouds.

### 4.3 Ablation Study

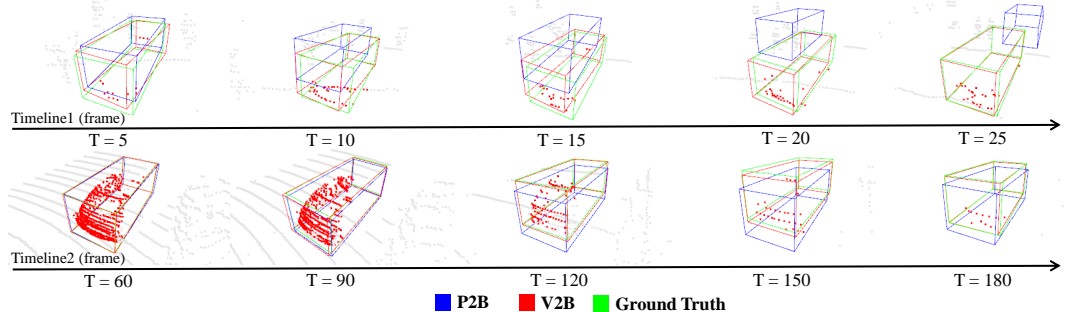

Figure 4: Visualization results of sparse (the first-row sequence) and dense (the second-row sequence) scenes on the car category. The green boxes are ground truth bounding boxes. The red boxes are the objects tracked by our V2B, while the blue boxes are the objects tracked by P2B. In addition, we mark the points of cars to red for better identification.

**Template feature embedding.** We study the impact of template feature embedding on tracking performance. As shown in Tab. 3, we report the results of the car category in the KITTI dataset. It can be seen that without using the template feature embedding (dubbed "w/o template feature"), the performance will be greatly reduced from 70.5 / 81.3 to 63.9 / 73.9 by a large margin. In addition, only using the local branch or global branch cannot achieve the best performance. Since template feature embedding builds the relationship between the template and search area, it will contribute to identify the potential target from the background in the search area. Therefore, when the template feature embedding is absent, the performance will be greatly reduced, which further demonstrates the effectiveness of the proposed template feature embedding for improving tracking performance.

Table 3: The ablation study results of different components on the car category.

| Module | *Success/Precision* |
|---|---|
| w/o template feature | 63.9 / 73.9 |
| local temple | 68.0 / 79.2 |
| global temple | 68.8 / 80.0 |
| w/o shape information | 67.6 / 78.2 |
| local geometric | 68.6 / 79.3 |
| global shape | 69.6 / 80.3 |
| default setting | **70.5 / 81.3** |

**Shape-aware feature learning.** For sparse point clouds, we further introduce shape-aware feature learning to enhance the ability to distinguish the potential target from the background in the search area. As shown in Tab. 3, we conduct experiments to demonstrate the effectiveness of the shape information. It can be seen from the table that without using shape-aware feature learning module (dubbed "w/o shape information"), the performance will reduce from 70.5 / 81.3 to 67.6 / 78.2. In addition, only using the local geometric branch or global shape branch cannot achieve the best performance. Since the shape generation network can capture 3D shape information of the object to learn the discriminative features of potential target so that it can be identified from the search area.

**Voxel-to-BEV target localization.** Different from SC3D [21] and P2B [52], our V2B adopts another route to localize potential target in object tracking. SC3D performs matching between the template and the exhaustive candidate 3D proposals to select the most similar proposal as the target. P2B and BAT use VoteNet [48] to generate 3D target proposals, and select the proposal with the highest score as the target. However, when facing sparse point clouds, it is hard to generate high-quality

Table 4: Comparison of different detection schemes on the **sparse** scenarios.

| Module | VoteNet | Voxel-to-BEV |
|---|---|---|
| Car | 56.9 / 72.0 | **64.7 / 77.4** |
| Pedestrian | 35.3 / 62.1 | **50.8 / 74.2** |
| Van | 30.7 / 39.0 | **46.8 / 55.1** |
| Cyclist | 23.9 / 30.0 | **30.4 / 37.2** |

proposals, so these methods may not be able to track the object effectively. Our V2B is an anchor-free method that does not require generating numerous 3D proposals. Therefore, our method can overcome the above concern. In order to prove this, we use VoteNet instead of voxel-to-BEV target localization to conduct experiments in the KITTI dataset. In Tab. 4, we report the results of different detection methods in the sparse scenarios. Likewise, we filter out sparse scenes in the test set for

evaluation according to the number of points (refer to the setting of Tab. 2). It can be found that the results of VoteNet are lower than that of voxel-to-BEV target localization, which further demonstrates the effectiveness of our method in sparse point clouds.

**Different voxel sizes.** We compress the voxelized point cloud into a BEV feature map for subsequent target center detection. Since the scope of object tracking is a large area, the size of voxel will affect the size of BEV feature map, thereby affecting the tracking performance. We hence study the impact of different voxel sizes on the tracking performance. Specifically, we consider four sizes, including 0.1, 0.2, 0.3, and 0.4 meters. The *Success/Precision* results of the four sizes are 52.5 / 62.2 (0.1m), 68.1 / 79.4 (0.2m), **70.5 / 81.3** (0.3m), and 69.1 / 80.0 (0.4m), respectively. When the voxel size is set to 0.3 meters, we achieve the best performance. A larger voxel size will increase the sparsity and cause the loss of the target's details. A smaller voxel size will increase the size of the BEV feature map, thereby increasing the difficulty of detecting the center.

**Template generation scheme.** Following [21, 52, 83], we study the impact of different template generation schemes on tracking performance. As shown in Tab. 5, we report the results of four schemes on the car category in the KITTI dataset. It can be seen from the table that our V2B outperforms SC3D, P2B, and BAT in all schemes by a large margin. Compared with these methods, our V2B can yield stable results on four template generation schemes, which further demonstrates that our method can consistently generate accurate tracking results in all types of frames.

Table 5: The results of different template generation schemes of different methods in the car category.

| Scheme | SC3D [21] | P2B [52] | BAT [83] | V2B (ours) |
|---|---|---|---|---|
| The First GT | 31.6 / 44.4 | 46.7 / 59.7 | 51.8 / 65.5 | **67.8 / 79.3** |
| Previous result | 25.7 / 35.1 | 53.1 / 68.9 | 59.2 / 75.6 | **70.0 / 81.3** |
| The First GT & Previous result | 34.9 / 49.8 | 56.2 / 72.8 | 60.5 / 77.7 | **70.5 / 81.3** |
| All previous results | 41.3 / 57.9 | 51.4 / 66.8 | 55.8 / 71.4 | **69.8 / 81.2** |

## 5 Conclusion

In this paper, we proposed a Siamese voxel-to-BEV (V2B) tracker for 3D single object tracking on sparse point clouds. In order to learn the dense geometric features of the potential target in the search area, we developed a Siamese shape-aware feature learning network that utilizes the target completion model to generate the dense and complete targets. In order to avoid using the low-quality proposals on sparse point clouds for target center prediction, we developed a simple yet effective voxel-to-BEV target localization network that can directly regress the center of the potential target from the dense BEV feature map without any proposal. Rich experiments on the KITTI and nuScenes datasets have demonstrated the effectiveness of our method on sparse point clouds.

## Acknowledgments

This work was supported by the National Science Fund of China (Grant Nos. U1713208, 61876084).

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
