# Appendix for "3D Siamese Voxel-to-BEV Tracker for Sparse Point Clouds"

## A  Overview

This supplementary material provides more details on network architecture, implementation, and experiments in the main paper to validate and analyze our proposed method.

In Sec. B, we provide specific network architecture and more details about the target center parameterization for the voxel-to-BEV target localization network. In Sec. C, we provide more details about template and search area generation in training and testing. In Sec. D, we present more experimental results, including quantitative results, visualization, and ablation study.

**Voxel-to-BEV Target Localization Network**

Figure 1: The architecture of our voxel-to-BEV target localization network.

## B  Network Architecture

In this section, we provide a specific network architecture used for the voxel-to-BEV target localization network. As shown in Fig. 1, we illustrate the specific network structure. Specifically, we first use the 3D network to aggregate features in the volumetric space. Then, we present the 2D network to aggregate features in the BEV space. After that, we introduce the center network for target localization. Finally, we provide more details on the target center parameterization.

**3D network.** We use a stack of 3D convolutions to the volumetric space to aggregate the features so that target's voxel can obtain rich target information. As shown on the left side of Fig. 1, we depict the specific structure of the network. Specifically, it consists of four 3D convolutions with the filter sizes $3 \times 3 \times 3$. In order to reduce memory consumption, we set the stride of four 3D convolutions to $1 \times 1 \times 2$, $1 \times 1 \times 1$, $1 \times 1 \times 2$, and $1 \times 1 \times 1$, respectively. Note that the stride along the $z$-axis is 2, so the feature size in the $x$-$y$ plane will not change. Finally, the 3D network outputs a new feature map with a size of $H \times W \times Z \times 2C$.

**2D network.** After projecting the voxelized point cloud into the bird's eye view (BEV) space through the max pooling function, we obtain a new BEV feature map with a size of $H \times W \times 2C$. We use a stack of 2D convolutions to construct a shallow encoder-decoder neural network to aggregate the features so that target's pixel can obtain rich target information. We show the specific network structure in the middle part of Fig. 1. Specifically, we first adopt three 2D convolutions (of filter sizes $3 \times 3$) and one transposed convolution (of filter size $2 \times 2$). And the strides of four 2D convolutions

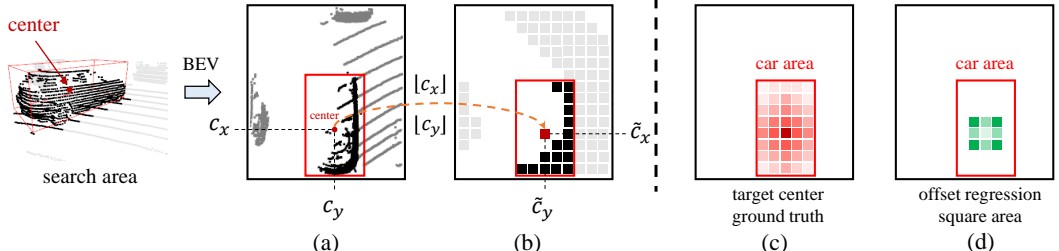

Figure 2: The details of the target center parameterization. (a) represents the continuous coordinates in the BEV space and (b) represents the discrete 2D grid in the BEV space. (c) is the target center ground truth and (d) is the square area used in the offset regression branch.

are 2×2, 1×1, 1×1, and 2×2, respectively. Note that the last 2D convolution is the transposed convolution. We also use a concatenated skip connection to fuse low-level and high-level features. Finally, after using a 2D convolution, we obtain a new feature map with a size of $H \times W \times 2C$.

**Center network.** Since we have known the size (length, width, and height) of the object in the template, we only need to regress the target center, offset, and rotation. As shown on the right side of Fig. 1, we illustrate the specific network structure. Specifically, we use three heads to regress the 2D center, offset & rotation, and $z$-axis location, respectively. For each head, we use two convolutions with filter sizes of 3×3 and 1×1. The output sizes are $H \times W \times 1$ (2D-center head), $H \times W \times 3$ (offset & rotation head) and $H \times W \times 1$ ($z$-axis head), respectively. Note that for the offset & rotation head, 3-dim means the 2-dim coordinate offset and 1-dim rotation.

**Target center parameterization.** To obtain the target center in the BEV space, we perform target center parameterization. Assuming the voxel size $v$ and the range of the search area $[(x_{min}, x_{max}), (y_{min}, y_{max})]$ in the $x$-$y$ plane, we can obtain the resolution of the BEV feature map by:

$$H = \lfloor \frac{x_{max} - x_{min}}{v} \rfloor + 1, W = \lfloor \frac{y_{max} - y_{min}}{v} \rfloor + 1 \tag{1}$$

where $\lfloor \cdot \rfloor$ is the floor operation. Given a 3D center $(x, y, z)$ of the target ground truth, we can compute the 2D target center $c = (c_x, c_y)$ in the $x$-$y$ plane by:

$$c_x = \frac{x - x_{min}}{v}, c_y = \frac{y - y_{min}}{v} \tag{2}$$

As shown in Fig. 2(a), we highlight the target center $(c_x, c_y)$ with a red dot. Note that the current target center is a continuous coordinate. After that, we perform floor operation to obtain the discrete target center $\tilde{c} = (\tilde{c}_x, \tilde{c}_y)$, which is given by

$$\tilde{c}_x = \lfloor c_x \rfloor, \tilde{c}_y = \lfloor c_y \rfloor \tag{3}$$

In Fig. 2(b), we highlight the discrete target center $(\tilde{c}_x, \tilde{c}_y)$ with a red grid. In this way, we can obtain the 2D grid of the search area.

For the 2D-center head, we follow [1] and generate the target center's ground truth $\mathcal{H} \in \mathbb{R}^{H \times W \times 1}$ based on the discrete 2D grid. Specifically, for each pixel $(i, j)$ in the 2D bounding box (refer to Fig. 2(c)), $\mathcal{H}_{ij}$ is defined by:

$$\mathcal{H}_{ij} = \frac{1}{d+1} \tag{4}$$

where $d$ represents the Euclidean distance between the pixel $(i, j)$ and the target center $(\tilde{c}_x, \tilde{c}_y)$. If $i = \tilde{c}_x$ and $j = \tilde{c}_y$, then $\mathcal{H}_{ij} = 1$. Besides, for any pixel outside the 2D bounding box, $\mathcal{H}_{ij}$ is set to 0. During training, we enforce the generated map $\hat{\mathcal{H}} \in \mathbb{R}^{H \times W \times 1}$ (refer to the 2D-center head in Fig. 1) to approach the ground truth $\mathcal{H}$ by using Focal loss [2], which is given by:

$$\mathcal{L}_{center} = -\sum \mathbb{I}[\mathcal{H}_{ij} = 1] \cdot (1 - \hat{\mathcal{H}}_{ij})^\alpha \log(\hat{\mathcal{H}}_{ij}) + \mathbb{I}[\mathcal{H}_{ij} \neq 1] \cdot (1 - \mathcal{H}_{ij})^\beta (\hat{\mathcal{H}}_{ij})^\alpha \log(1 - \hat{\mathcal{H}}_{ij}) \tag{5}$$

where $\mathbb{I}(cond.)$ is the indicator function. If $cond.$ is true, then $\mathbb{I}(cond.) = 1$, otherwise 0. By minimizing the $\mathcal{L}_{center}$, it is desired that the 2D discrete center can be effectively detected. We empirically set $\alpha = 2$ and $\beta = 4$ in all experiments.

For the offset & rotation head, we regress the offset of the continuous ground truth 2D center. Specifically, we consider a square area with radius $r$ (refer to Fig. 2(d)) around the object center to improve the accuracy of the offset regression. Note that here we also add rotation regression.

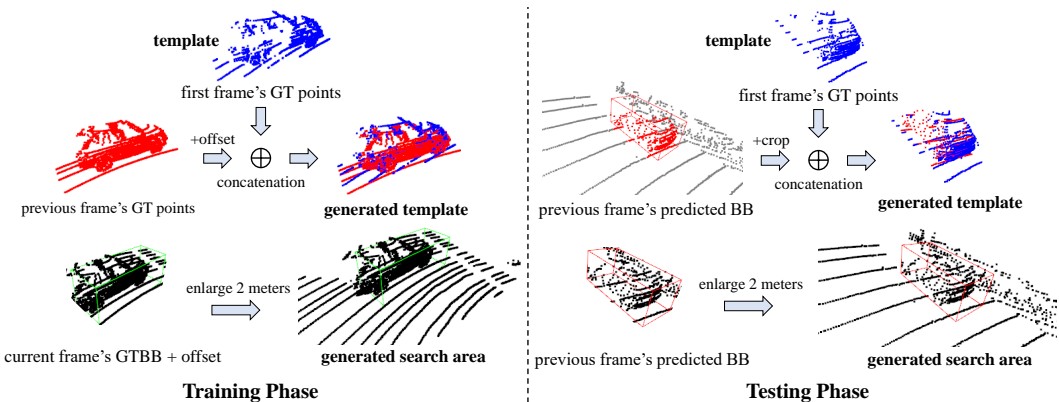

Figure 3: The processing of the template and search area in the training phase and the testing phase, respectively. "BB" denotes the bounding box, and "GT" denotes the ground truth.

Assuming a generated map $\hat{\mathcal{O}} \in \mathbb{R}^{H \times W \times 3}$ (refer to the offset & rotation head in Fig. 1), the error of the offset and rotation is formulated as:

$$\mathcal{L}_{off} = \sum_{\triangle x=-r}^{r} \sum_{\triangle y=-r}^{r} \left| \hat{\mathcal{O}}_{\tilde{c}+(\triangle x, \triangle y)} - [c - \tilde{c} + (\triangle x, \triangle y), \theta] \right| \tag{6}$$

where $\tilde{c}$ and $c$ mean the discrete and continuous position of the ground truth center, respectively. Besides, $\theta$ indicates the ground truth rotation angle and $[\cdot, \cdot]$ is the concatenation operation. We empirically set $r = 2$ in all experiments.

For the $z$-axis head, we directly regress the $z$-axis position of the target center from the BEV feature map. Assuming the generated feature map $\hat{\mathcal{Z}} \in \mathbb{R}^{H \times W \times 1}$, the error of the $z$-axis position is written as:

$$\mathcal{L}_z = \left| \hat{\mathcal{Z}}_{\tilde{c}} - z \right| \tag{7}$$

where $\tilde{c}$ is the discrete target center, and $z$ is $z$-axis center's ground truth.

To generate the predicted target center, we first select the position with the highest response on the feature map $\hat{\mathcal{H}} \in \mathbb{R}^{H \times W \times 1}$ as the 2D center on the $x$-$y$ plane. Assuming the selected position is $(i, j)$, we then obtain the value $\hat{\mathcal{O}}_{ij} \in \mathbb{R}^3$ from the feature map $\hat{\mathcal{O}} \in \mathbb{R}^{H \times W \times 3}$ as the value of offset and rotation. After that, we can obtain the continuous target center $(t_x, t_y)$ by $t_x = i + \hat{\mathcal{O}}_{ij,0}$, $t_y = j + \hat{\mathcal{O}}_{ij,1}$, where $\hat{\mathcal{O}}_{ij,0}$ and $\hat{\mathcal{O}}_{ij,1}$ are the offsets of the $x$-axis and the $y$-axis. Note that the rotation value $\hat{\mathcal{O}}_{ij,2}$ is performed on the final bounding box. In addition, we obtain the value $\hat{\mathcal{Z}}_{ij}$ from the feature map $\hat{\mathcal{Z}} \in \mathbb{R}^{H \times W \times 1}$ as the target's $z$-axis center.

## C  Implementation Details

**Template and search area in training.** Following [3, 4], we adopt the same strategy to generate the template and search area during training. For the current frame, the template is generated by fusing two frames, *i.e.*, the first frame and the previous frame (if exists). As shown in the left half of Fig. 3, we combine the points inside the first frame's ground truth bounding box (GTBB) and the points inside the previous frames' GTBB plus the random offset as the template of the current frame. During training, we sample 512 points from the template by discarding and duplicating the points. For the search area, we enlarge the current frame's GTBB by 2 meters plus the random offset. Likewise, we sample 1024 points from the search area by discarding and duplicating the points.

**Template and search area in testing.** In the right half of Fig. 3, we show the specific process of generating the template and search area. For testing, we fuse the points inside the first frame's ground truth bounding box (GTBB) and the previous result's point cloud (if exists) as the template of the current frame. For the search area, we first enlarge the previously predicted bounding box by 2 meters in the current frame and then collect the points lying in it to construct the search area. Note that, unlike the training phase, we do not apply random offset augmentation to the previous result's point

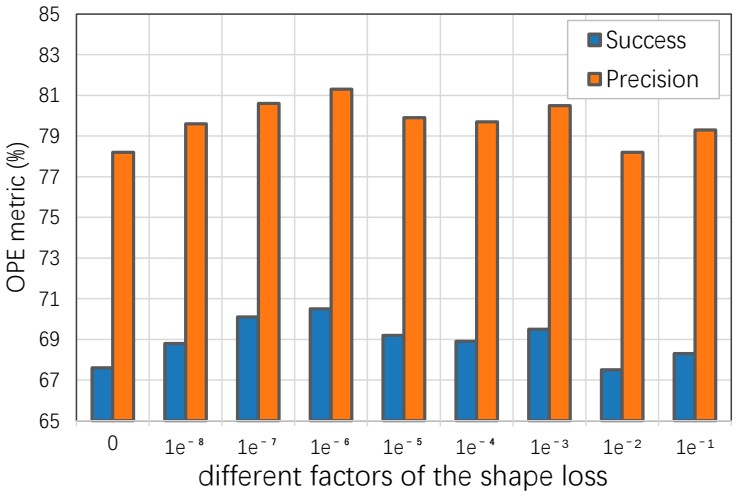

Figure 4: Ablation study results (*Success* and *Precision*) of different factors of the shape generation.

cloud. Besides, we sample 512 points in the template and 1024 points in the search area during testing.

**Template generation schemes.** In the experiment, we also report the tracking performance of four different template generation schemes. They are dubbed as "The First GT", "Previous result", "The First GT & Previous result", and "All previous results", respectively. "First GT" means that we only use the first frame as the template, and "Previous result" means that we only use the tracked result's point cloud in the previous frame as the template. Thus, "The First GT & Previous result" is a combination of "First GT" and "Previous result". Besides, "All previous results" represents that we align all of the previous tracked results' point clouds as a template.

## D  Experiments

**Different factors of the shape loss.** As shown in Fig. 4, we study the impact of different factors of the shape loss on the tracking performance. It can be seen that when the factor is set to $10^{-6}$, we achieve the best performance. Furthermore, according to the figure, different factors are insensitive to the performance.

**Quantitative results.** To better validate and analyze the proposed method, we also show the performance of different objects in different point intervals. Specifically, we divide the interval according to the number of points lying in the ground truth bounding boxes in the test set. For large-size categories such as car and van, we set four intervals, including [0, 150), [150, 1000), [1000, 2500), and [2500, +∞). For small-size categories such as pedestrian and cyclist, we set four intervals, including [0, 100), [100, 500), [500, 1000), and [1000, +∞). As shown in Tab. 1, we report the *Success* and *Precision* of SC3D [3], P2B [4], BAT [5], and our V2B. It can be seen that our method is superior to other methods in terms of the mean of the four categories.

**Visualization.** As shown in Fig. 5, we provide more visualization results of our method for four categories, including car, pedestrian, van, and cyclist. It can be seen that our method can accurately localize targets in both dense and sparse scenes. As shown in Fig. 6, we also compare the tracking results of SC3D [3], P2B [4], and our method.

**Failure cases.** As shown in Fig. 7, we provide the visualization results of the failure cases. It can be found that our method will fail in extremely sparse scenes. If the tracking fails in the previous frame, a poor-quality search area will be generated for the current frame, affecting the tracking results of subsequent frames.

Table 1: The results of *Success*/*Precision* of different methods at different point intervals. "Mean" represents the average results of four categories.

| Method | Car | Pedestrian | Van | Cyclist | Mean |
|---|---|---|---|---|---|
| Total Frame Number | 6424 | 6088 | 1248 | 308 | 14068 |
| Interval | [0, 150) | [0, 100) | [0, 150) | [0, 100) | |
| Frame Number | 3293 | 1654 | 734 | 59 | 5740 |
| SC3D [3] | 37.9 / 53.0 | 20.1 / 42.0 | 36.2 / 48.7 | **50.2 / 69.2** | 32.7 / 49.4 |
| P2B [4] | 56.0 / 70.6 | 33.1 / 58.2 | 41.1 / 46.3 | 24.1 / 28.3 | 47.2 / 63.5 |
| BAT [5] | 60.7 / 75.5 | 48.3 / **77.1** | 41.5 / 47.4 | 25.3 / 30.5 | 54.3 / 71.9 |
| V2B (ours) | **64.7 / 77.4** | **50.8** / 74.2 | **46.8 / 55.1** | 30.4 / 37.2 | **58.0 / 73.2** |
| Interval | [150, 1000) | [100, 500) | [150, 1000) | [100, 500) | |
| Frame Number | 2156 | 3112 | 333 | 145 | 5746 |
| SC3D [3] | 36.1 / 53.1 | 17.7 / 38.2 | 38.1 / 53.3 | **44.7 / 76.0** | 26.5 / 45.6 |
| P2B [4] | 62.3 / 78.6 | 25.1 / 46.0 | 41.7 / 50.5 | 35.4 / 46.5 | 40.3 / 58.5 |
| BAT [5] | 71.8 / 83.9 | 45.0 / 71.2 | 44.0 / 51.6 | 41.5 / 52.2 | 54.8 / 74.3 |
| V2B (ours) | **77.5 / 87.1** | **46.8 / 72.0** | **51.2 / 59.6** | 44.4 / 53.9 | **58.5 / 76.5** |
| Interval | [1000,2500) | [500,1000) | [1000,2500) | [500,1000) | |
| Frame Number | 693 | 1071 | 78 | 42 | 1884 |
| SC3D [3] | 33.8 / 48.7 | 15.0 / 37.1 | 35.9 / 50.3 | 34.9 / **69.5** | 23.2 / 42.6 |
| P2B [4] | 51.9 / 68.1 | 28.4 / 49.9 | 40.7 / 49.7 | 25.7 / 37.7 | 37.5 / 56.3 |
| BAT [5] | 69.1 / 81.0 | 35.2 / 61.7 | 50.3 / 61.3 | 34.9 / 48.7 | 48.3 / 68.5 |
| V2B (ours) | **72.3 / 81.5** | **47.2 / 74.3** | **61.3 / 67.8** | **42.3** / 52.0 | **56.9 / 76.2** |
| Interval | [2500,+∞) | [1000,+∞) | [2500,+∞) | [1000,+∞) | |
| Frame Number | 282 | 251 | 103 | 62 | 698 |
| SC3D [3] | 23.7 / 35.3 | 14.5 / 35.3 | 30.5 / 42.4 | 27.7 / **64.2** | 21.8 / 38.9 |
| P2B [4] | 43.8 / 61.8 | 27.1 / 49.1 | 33.8 / 39.7 | 24.6 / 34.2 | 34.6 / 51.5 |
| BAT [5] | 61.6 / 72.9 | 32.6 / 58.6 | 48.2 / 57.9 | 26.7 / 37.9 | 46.1 / 62.4 |
| V2B (ours) | **82.2 / 90.1** | **53.8 / 82.6** | **60.9 / 65.9** | **41.2** / 50.4 | **65.2 / 80.3** |

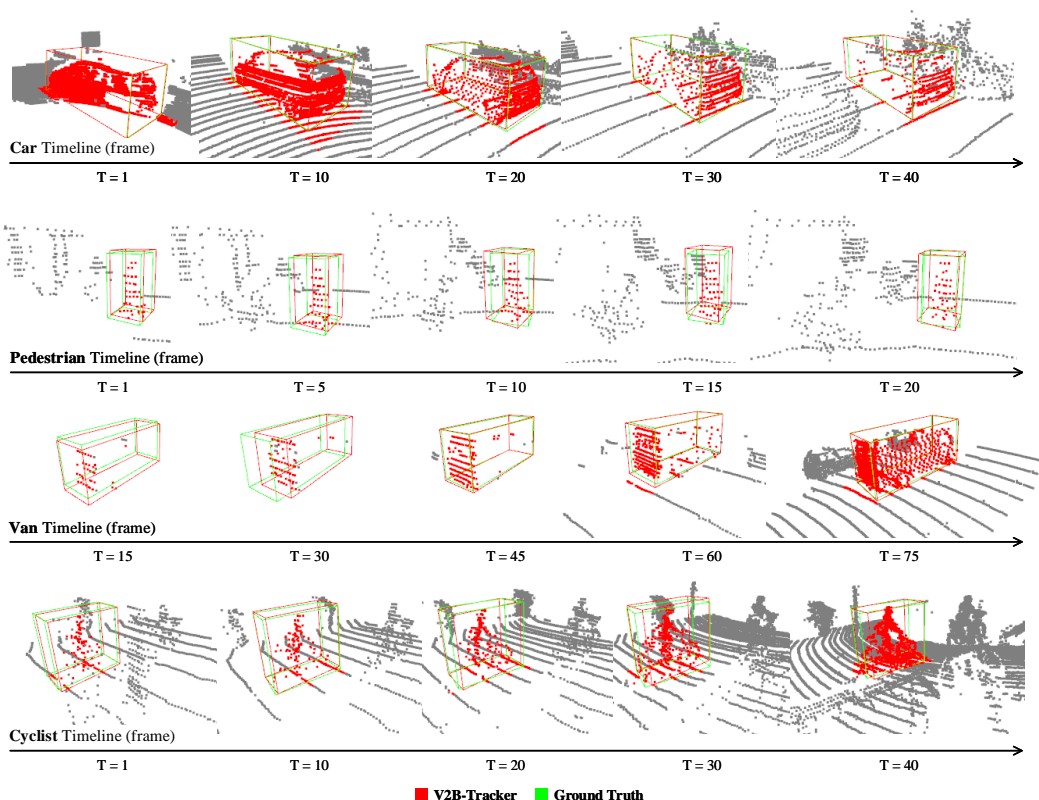

Figure 5: Visualization results of our method. From top to bottom, the visualization results are cars, pedestrians, vans, and cyclists, respectively. The green boxes are ground truth bounding boxes, and the red boxes are the predicted bounding boxes of our V2B. Note that we mark the points of the ground truth in red for better identification.

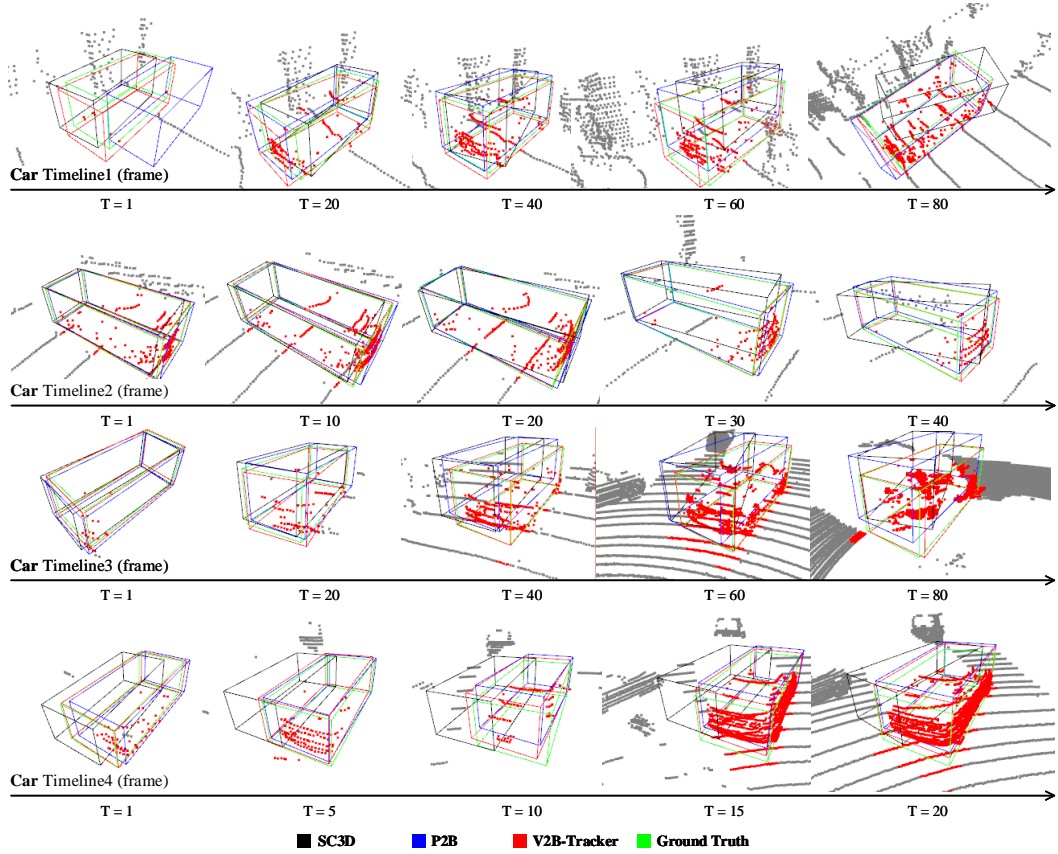

Figure 6: Visualization results of SC3D [3], P2B [4], and our method on the car category. The green boxes are ground truth bounding boxes. The **black**, blue, red boxes are the predicted bounding boxes of SC3D, P2B, and our V2B, respectively. Note that we mark the points of the ground truth in red for better identification.

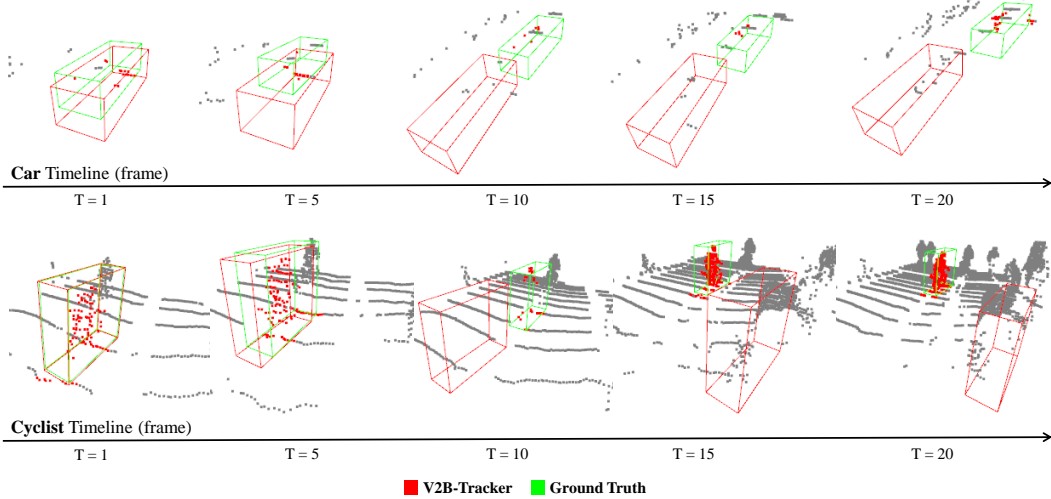

Figure 7: Visualization results of failure cases. The first-row sequence is the result of the car category, and the second-row sequence is the result of the cyclist category. The green boxes are ground truth bounding boxes, and the red boxes are the predicted bounding boxes of our V2B. Note that we mark the points of the ground truth in red for better identification.