# OpenReview forum: "3D Siamese Voxel-to-BEV Tracker for Sparse Point Clouds"
_NeurIPS.cc/2021/Conference — NeurIPS 2021 Poster_

### Official Review · Reviewer_yABo · 2021-07-06

**Rating:** 6
**Confidence:** 4

**Summary:**

this paper proposed a Siamese voxel-to BEV tracker, including global and local template matching, shape-completion network and a voxel-to-BEV target localization network.

**Limitations And Societal Impact:**

yes

**Main Review:**

The results  look promising.

My major concern if novelty is limited.  The proposed method is an ensemble of well-known components: template matching (eg. SiamRPN[1]), shape-completion and bev localization (e.g. LaserNet[2]).

Ablation studies of global and local template features are desired.

[1]Li, Bo, et al. "High performance visual tracking with siamese region proposal network." Proceedings of the IEEE conference on computer vision and pattern recognition. 2018.
[2]Meyer, Gregory P., et al. "Lasernet: An efficient probabilistic 3d object detector for autonomous driving." Proceedings of the IEEE/CVF Conference on Computer Vision and Pattern Recognition. 2019.

**Time Spent Reviewing:**

3

---

> ### Author Response · Authors · 2021-08-12
> **Response to Reviewer yABo**
>
> We thank the reviewer for your valuable comments. Our responses to the comments on ablation study and novelty are as below:
>
> **Q1**: Ablation studies of both local and global features in template feature embedding module.
>
> **A1**: We list the ablation study results of different settings as follows:
>
> |&emsp;&emsp;Settings&emsp;&emsp;| &emsp;&ensp;Car | Pedestrian| &emsp;&ensp;Van | &ensp;&nbsp;Cyclist |
> |:-:|:-:|:-:|:-:|:-:|
> | local feature | 68.0 / 79.2 | 47.2 / 71.5 | 46.3 / 56.1 | 38.6 / 47.2 |
> | global feature | 68.8 / 80.0 | 47.9 / 72.2 | 47.5 / 57.1 | 39.2 / 48.0 |
> | both | **70.5** / **81.3** | **48.3** / **73.5** | **50.1** / **58.0** | **40.8** / **49.7** |
>
> It can be seen that when simultaneously using local and global features, our method achieves the best results on all four categories.
>
> &emsp;
>
>
> **Q2**: The proposed method is an ensemble of well-known components such as SiamRPN and LaserNet.
>
> **A2**: Due to the sparsity and irregularity of point clouds, 2D single object tracking methods such as SiamRPN cannot be directly applied to 3D single object tracking. Although our method and SiamRPN use the Siamese network in single object tracking, due to the huge gap between 2D images and 3D point clouds, they are different in feature extraction. In our method, we first employ template feature embedding with the similar Siamese backbone network to encode the search area. In order to track the object with sparse point clouds, we then develop the target completion model (shape-aware feature learning module) to learn dense geometric features of the potential target in the search area so that the dense and complete point clouds of the potential target can be generated. Specifically, in the target completion model, the global-local branch is constructed with the gate mechanism to enhance the feature of the potential target and suppress the background in the search area.
>
> &emsp;In our method, the localization network is also different from 3D object detection methods such as LaserNet. LaserNet is the proposal based 3D detection method on the range view representation. However, due to the non-uniform and few points in the sweep, the range view map obtained from the sparse point clouds cannot provide enough local geometric information of the object, and thus cannot generate high-quality proposals for the object in the sparse point clouds. In order to avoid using the low-quality proposals on sparse point clouds for target center prediction, we develop a simple yet effective target center localization model without any proposal. The 3D center of the potential target can be directly regressed with the highest response in the dense BEV feature map, where the dense BEV feature map is generated by voxelizing the learned dense geometric features and performing max-pooling along the z axis. Thus, with the constructed dense BEV feature map, for sparse point clouds, our method can more accurately localize the target center.

---

> > ### Comment · Reviewer_yABo · 2021-08-25
> > **Thanks for the response**
> >
> > Thanks the authors for the response and additional ablation studies provided. However I am still not convinced the novelty is substantial. BEV proposal-free localization is not novel:
> > Object as Hotspots: An Anchor-Free 3D Object Detection Approach via Firing of Hotspots, ECCV 2020
> >
> > Center-based 3D Object Detection and Tracking, CVPR 2021 (I know CVPR 2021 should be treated as concurrent work but this one has been on arxiv for a long time).
> >
> > AFDet: Anchor Free One Stage 3D Object Detection, CVPRW 2020

---

> > > ### Author Response · Authors · 2021-08-27
> > > **Response to Reviewer yABo**
> > >
> > > We thank the reviewer for your reply. Although our localization network and the mentioned 3D detection methods are anchor-free methods, the generation of the BEV feature map is different. In our method, the BEV feature map is generated from the dense geometric features of point clouds learned by the target completion model (our shape-aware feature learning module), where the dense and complete point clouds of the potential target in the search area can be obtained. However, in 3D detection methods such as 3D CenterPoint, the BEV feature map is generated from the sparse features of point clouds, where the sparse features are learned by using simple PointNet on voxels in the sparse volume space. In addition, our method uses the simple and efficient max pooling along the z-axis on the dense geometric features to generate the **dense BEV feature map**, while 3D detection methods such as 3D CenterPoint apply 3D convolution on the sparse features in the volume space to generate the **sparse BEV feature map** due to large amounts of empty voxels. Compared to the sparse BEV map, the dense BEV feature map can more accurately localize the 3D center of the target, where the low responses in the voxelized feature map can be suppressed better by performing max pooling along the z-axis. We conduct the tracking experiments with 3D CenterPoint. The quantitative results are as follows:
> > >
> > > | &emsp;&emsp;&ensp;Method | Success/Precision |
> > > | ----| ----|
> > > | &ensp;&ensp;3D CenterPoint | &ensp;&emsp;65.3 / 75.4 |
> > > | &ensp;V2B-Tracker (ours) | &ensp;&emsp;**70.5** / **81.3** |
> > >
> > > It can be seen that our method obtains better performance than 3D CenterPoint. We will add the results of 3D CenterPoint in the revised manuscript.

---

### Official Review · Reviewer_kZCW · 2021-07-15

**Rating:** 6
**Confidence:** 4

**Summary:**

This paper proposes a point cloud-based 3D single-object tracking method. To improve tracking performance in sparse point cloud scenarios, the proposed method incorporates the shape completion technique to enhance feature learning. Results are evaluated on the KITTI dataset demonstrating that the proposed method outperforms baselines

**Limitations And Societal Impact:**

Limitation is addressed in the paper and negative societal impact is not discussed which I think is okay

**Main Review:**

Paper Strengths

1. The engineering efforts on designing local and global feature embedding for template feature extraction are intuitive, which also significantly improve performance as shown in Table 4 ablation experiments

2. Qualitative analysis on results for sparse point cloud intuitively demonstrating the usefulness of the proposed method

Paper Weaknesses

1. It is still vague to me if the main contributions of this paper are very useful through current experiments:

a) Regarding shape completion, the main difference claimed in this paper compared to prior work is that the proposed method performs shape completion in the search area while [13] performs shape completion for the template. Although the ablation experiments in Table 4 shows that without shape information by completion, the performance drops, it also needs to compare with shape completion on the template as [13]. As a result, I am not sure if the proposed idea is better or worse than [13]. Also, the absolute performance comparison in Table 1 does not help answer this question because presumably, the code base is different so there are lots of other factors affecting the overall performance. It would be easier to compare two ideas (this paper and [13]) by adding shape completion for the template in the proposed method, and/or change the shape completion to the search area or candidate shapes for [13]’s codebase. Also, intuitively it is unclear to me how performing shape completion for template or search area will make differences. Or maybe we should do both?

b) Regarding the voxel-to-BEV target localization, the proposed method essentially just converts the 3D voxel features into 2D by performing max pool on the z-axis. It is unclear to me, how will the feature max pool on the z-axis help the later network (z-axis head) to better localize the z-axis center? In general, people will think that max-pooling is throwing away information so intuitively it should make the z-axis center estimation harder using the maxpooled BEV features. This contradicts the paper’s claim (line 68) that after feature pool in the z-axis, the regression on both 2D center and z-axis center become more effective and accurate. Besides the above analysis, there is also no detailed analysis on this voxel-to-BEV ablation. For example, in addition to the final performance analysis, can we add an experiment showing that after adding or not adding vowel-to-BEV, will the performance on center estimation (both the 2D center and z-axis center) change significantly?

2. The network design is not justified. According to the paper (e.g., line 126) PointNet++ is extensively used in the method. Why not use more advanced point cloud processing methods since PointNet++ is 4 years old (NeurIPS 2017).

3. Also, it seems not reasonable to me that the evaluation (testing) is only performed on two sequences of KITTI? At this moment, even the entire KITTI tracking dataset is too small and cannot be used as a robust testbed for evaluation, let alone only two sequences of it. I understand that this paper is following prior work on this evaluation protocol, but I feel this protocol is not fully justified, in the cases that there are many better testbeds existing. For example, one can use nuScenes for more robust evaluation which has 150 sequences for validation and 150 sequences for testing with more objects. Under the current evaluation protocol, I am not confident any number or analysis we obtain from the only two KITTI sequences is robustly meaningful. Results on Cars might have some implications but results on Van and Cyclist should have a high variation because there are simply too few examples for those objects in two KITTI sequences. I would suggest that authors add one more dataset for evaluation and comparison with other methods, either nuScenes or Waymo for example, unless there is a strong reason that we can only use the two KITTI sequences for evaluation


**Time Spent Reviewing:**

2.5

---

> ### Author Response · Authors · 2021-08-12
> **Response to Reviewer kZCW**
>
> We thank the reviewer for your valuable comments. Our responses to the comments are as below:
>
> **Q1**: Differences between the shape completion of the template (SC3D [13]) and the search area (our method).
>
> **A1**: The main differences are two-fold. On the one hand, SC3D [13] uses a simple auto-encoder structure to learn the template completion model from the template samples (given bounding boxes of the targets), while our method is a complex point cloud generation method that learns the target completion model (shape-aware feature learning module) from the samples of search areas with the gate mechanism to enhance the feature of the potential target and suppress the background in the search area. Due to limited templates and large variations of the potential target in the search area, the template completion model in SC3D cannot recover complex geometric structures of the potential target well, which might lead to the inaccurate matching. On the other hand, the template completion model in SC3D only employs PointNet to extract point features of sparse point clouds, while our target completion model constructs a global-local branch to extract global shape features and local geometric features of sparse point clouds. We list the ablation study of shape completion as follows:
>
> | &emsp;&emsp;&emsp;&emsp;&emsp;V2B-Tracker &emsp;&emsp;| &emsp;&ensp;Car | Pedestrian| &emsp;&ensp;Van | &ensp;&nbsp;Cyclist |
> |:-:|:-:|:-:|:-:|:-:|
> | shape completion in SC3D| 67.3 / 78.4 | 46.1 / 70.2 | 47.5 / 56.6 | 36.9 / 45.4 |
> | our shape-aware feature learning | **70.5** / **81.3** | **48.3** / **73.5** | **50.1** / **58.0** | **40.8** / **49.7** |
>
> It can be seen that our shape completion model outperforms the completion model in SC3D on all four categories, where template shape completion in SC3D is used in our framework.
>
> &emsp;
>
> **Q2**: How will the feature max pooling on the z-axis help the later network (z-axis head) to better localize the z-axis center?
>
> **A2**: In our localization network we regress the target center from the highest response in the learned feature map, where the point with the highest response is regarded as the predicted center. In the sparse volume space, due to the large number of empty voxels, the differences between the responses in the voxelized feature map might not be remarkable. Thus, the highest response in the feature map is difficult to distinguish from the low responses, leading to the inaccurate regression of the 3D center of the target, including the z-axis center. By performing max-pooling on the voxelized feature map along the z-axis, we can obtain the dense BEV feature map, where the low responses in the voxelized feature map can be suppressed. Thus, compared to the voxelized feature map, we can more accurately localize the 2D center of the target with the highest response in the dense BEV feature map. The response of the 2D center (i.e., max-pooling feature along the z-axis) in the BEV feature map actually contains the geometric structure information of the potential target while the responses of other points in the BEV feature map do not. Therefore, we can more accurately regress the z-axis center of the target with the max-pooling feature along the z-axis.
>
> &emsp;
>
> **Q3**: Tracking results without voxel-to-BEV.
>
> **A3**: We conduct the experiment by directly using the voxelized feature map in the volume space instead of the BEV map for target localization. The results are listed as follows:
>
> |&emsp;&ensp;V2B-Tracker|&emsp;&ensp;Car|Pedestrian|&emsp;&ensp;Van|&ensp;&nbsp;Cyclist|
> |:-:|:-:|:-:|:-:|:-:|
> | volume space| 68.3 / 78.9 | 46.5 / 71.0 | 48.7 / 56.2 | 38.4 / 47.5 |
> |Voxel-to-BEV (ours)| **70.5** / **81.3** | **48.3** / **73.5** | **50.1** / **58.0** | **40.8** / **49.7** |
>
> It can be seen that our voxel-to-BEV structure can achieve better performance.
>
> &emsp;
>
> **Q4**: Why not use more advanced point cloud processing methods than PointNet++?
>
> **A4**: In order to reduce the inference time of the network, we just use PointNet++ as the backbone for feature extraction. It can be replaced with a powerful network such as PointWeb, KPConv, and RandLA-Net. We will clarify it in the revised manuscript.
>
> &emsp;
>
> **Q5**: Add the evaluation results on the nuScenes dataset.
>
> **A5**: Since the ground truth of the online test set of nuScenes for multiple object tracking cannot be obtained, we adopt the nuScenes training set (700 sequences) and validation set (150 sequences) for 3D single object tracking by generating a tracklet for each instance in the scenes. In the experiment, 700 sequences of the training set are used for training, and 150 sequences of the validation are used for testing. The results on the car and pedestrian categories are shown as follows:
>
> | &emsp;&emsp;&ensp;Method&ensp;&ensp;&ensp; |  &ensp;&ensp;&ensp;Car | Pedestrian |
> |:-:|:-:|:-:|
> |  SC3D [13]  |   37.3 / 43.8   | 13.6 / 33.2 |
> |  P2B [12] |   40.2 / 47.0   | 21.3 / 47.8 |
> |  V2B-Tracker (ours)  |   **46.0** / **54.8**   | **27.9** / **53.6** |
>
> It can be seen that our method can achieve better performance than the other two methods on the nuScenes dataset. We will add the results of the nuScenes dataset in the revised manuscript.

---

> > ### Comment · Reviewer_kZCW · 2021-08-26
> > **Post-rebuttal review**
> >
> > After reading other reviews and authors’ feedback, I would like to keep my original score. First of all, I appreciate the authors’ efforts in addressing all 7 reviewers’ comments, which requires significant effort. Regarding feedback to my concerns, I would like to acknowledge that my concerns have been partially addressed but not completely addressed (some have been overlooked), which is why I did not further increase my score.
> >
> > 1. Specifically, feedback on my Q1 is great, which shows that higher performance is due to a better share completion model. However, it is still unclear to me where we should use the shape completion model, should it be applied to the search area or template, or both? Would be better to have a direct comparison for the proposed method.
> >
> > 2. Explanation on Q2 is kind of convincing, however, results for Q3 are on the overall evaluation, while my original concern is trying to understand where the improvement comes from, i.e., a breakdown of the overall improvement. Is it really because the improvement is on 2D center and z-axis estimation? It would be more clear if the ablation is for the center estimation part and other parts instead of the overall performance so that we can understand better where the actual improvement comes from
> >
> > 3. Explanation on Q4 and results on Q5 is sufficient. It would be really great to have a new standard on bigger datasets such as the comparison shown in Q5 because I really feel like the evaluation on only 2 sequences of KITTI data is not robust.

---

> > > ### Author Response · Authors · 2021-08-30
> > > **Response to Reviewer kZCW**
> > >
> > > We are very grateful for your detailed responses. Our supplementary responses to **Q1** and **Q3** are as below:
> > >
> > > **Q1**: Where should we use the shape completion model, search area, or template, or both?
> > >
> > > **A1**: The proposed shape completion model (shape-aware feature learning module) is currently applied to the search area to generate dense point clouds. Nonetheless, our completion model can also be applied to the template or both, which corresponds to generate the dense template/potential target. We apply our model to the template, search area, both template and search area to show the tracking performance in different settings. The quantitative results are as follows:
> > >
> > > | &emsp;&emsp;&emsp;&ensp;V2B-Tracker &emsp;&emsp;&emsp;| &emsp;&ensp;Car&emsp; | &emsp;Pedestrian&emsp; | &emsp;&ensp;Van &emsp;| &emsp;&ensp;Cyclist&emsp; |
> > > |:-:|:-:|:-:|:-:|:-:|
> > > | template | 67.9 / 79.1 | 45.8 / 70.4 | 48.2 / 57.2 | 36.4 / 44.7 |
> > > | search area | 70.5 / 81.3 | 48.3 / 73.5 | 50.1 / 58.0 | 40.8 / 49.7 |
> > > | template + search area | **70.9** / **82.2** | **48.7** / **73.7** | **50.9** / **58.7** | **41.4** / **50.4** |
> > >
> > > Since the template shape completion only generates the dense template, it cannot effectively enhance the shape information of the potential target in the search area. Thus, the results with our shape completion model on the template is lower than those on the search area. When simultaneously using the shape completion model on the template and search area, we can enhance the shape information of the template and the potential target in the search area, respectively. Nonetheless, due to the limited gain with our completion model on the template, the tracking performance with our model on the template and search area is not significantly improved.
> > >
> > > &emsp;
> > >
> > > **Q3**: The breakdown of the overall improvement of the proposed method.
> > >
> > > **A3**: We conduct experiments of our method on the KITTI tracking dataset to show improvements of each part, including template feature embedding (template embedding), shape-aware feature learning (shape completion), and voxel-to-BEV localization (center estimation). According to our proposed framework, we define our baseline as the combination of PointNet++ and localization in the volume space (without max pooling along the z-axis) for 3D single object tracking. Based on the baseline, we sequentially add the template embedding (**TE**), shape completion (**SC**), and replace the localization in the volume space with the voxel-to-BEV localization (**BEV**) to show the gain of each part. The detailed quantitative results of each part are as follows:
> > >
> > > | &emsp;&emsp;&emsp;&emsp;&emsp;Methods&emsp;&emsp;&emsp;| &emsp;&ensp;Car&emsp; | &emsp;Pedestrian&emsp; | &emsp;&ensp;Van &emsp;| &emsp;&ensp;Cyclist&emsp; |
> > > |:-|:-:|:-:|:-:|:-:|
> > > | baseline | 53.2 / 66.1 | 24.3 / 43.2 | 37.3 / 45.8 | 29.3 / 41.9 |
> > > |baseline + TE | 61.3 / 73.9 | 33.8 / 54.4 | 42.8 / 50.1 | 33.1 / 45.9 |
> > > |baseline + TE + SC | 68.3 / 78.9 | 46.5 / 71.0 | 48.7 / 56.2 | 38.4 / 47.5 |
> > > |baseline + TE + SC + BEV | **70.5** / **81.3** | **48.3** / **73.5** | **50.1** / **58.0** | **40.8** / **49.7** |
> > >
> > > By comparing the results in adjacent rows, it can be seen that each part of our method can effectively improve the performance of the 3D single object tracking task.

---

### Official Review · Reviewer_CfXd · 2021-07-15

**Rating:** 6
**Confidence:** 4

**Summary:**

This paper introduced a siamese voxel2BEV object tracker for autonomous driving applications. The proposed network architecture is robust to sparse point cloud inputs, and can improve tracking performance significantly. Experimental results on KITTI benchmarks suggest that the proposed approach is highly effective.

**Limitations And Societal Impact:**

I have a broader impact question related to the overall goal of the project: I understand the scope of the project is 3D object tracking on LiDAR inputs. However for autonomous driving applications where safety is very important, why not use additional RGB information if necessary? My guess is that the current algorithm can be extended to deal with RGB+LiDAR inputs, so it'll be interesting if authors can provide more discussions about it.

**Main Review:**

Strength:
- The paper tackles a challenging task in autonomous driving, and the overall design of the algorithm is a sensible solution for this task.
- Experimental results on KITTI benchmarks are great. The method surpasses existing solutions by a fairly big margin.
- The presentation of the paper is relatively clear, and the visualizations in the paper are helpful for readers to grasp the intuitions of the design.
- Regardless, I hope the authors release the code as they promised in the supplementary material, especially since this is an application paper, and would be useful for future research projects.

Weakness:
- The worse performance on the cyclist category seems to suggest that there are limitations in the approach. The authors argue that this is due to the lack of training data.  Suppose there are not sufficient data in other categories, would the trend of the result still remain the same?
- Since this is an object tracking paper, it'll be helpful if authors can include videos that demonstrate the tracking outputs. Visualizations in the supplementary material are helpful but I think videos are more intuitive.
- Minor: is the color scheme in Fig. 3 mean anything? When I first saw this diagram, I tried mapping similar colors to certain sections but eventually failed.
- Minor: The paper is only tested on the KITTI dataset, and there are certain places in the whole pipeline that requires parameter tunings (such as coefficients for loss functions). What is the general guidance for parameter tuning when presented with new datasets? How do authors envision the proposed algorithm generalizes to new datasets or real autonomous driving scenarios?

**Time Spent Reviewing:**

5

---

> ### Author Response · Authors · 2021-08-12
> **Response to Reviewer CfXd**
>
> We thank the reviewer for the valuable comments. Our responses to your concerns are as below:
>
> **Q1**: Release the code.
>
> **A1**: We are sure that we will release the code on Github.
>
> &emsp;
>
> **Q2**: The worse performance on the cyclist category is due to lack of training data. If the data of the other three categories (car, pedestrian and van) are not sufficient, will the trend of the results of the other categories remain the same?
>
> **A2**: We conduct experiments on the other three categories under a small amount of training data. Specifically, for each category, we use the same number of training data as the cyclist category to train our network. The quantitative results are as follows:
>
> Table: Evaluation results of the car, pedestrian, and van categories with insufficient training data
>
> |&emsp;&emsp;Method| &emsp;Data | &emsp;&ensp;Car | Pedestrian | &emsp;&nbsp;Van |
> |:-:|:-:|:-:|:-:|:-:|
> | &emsp;V2B-Tracker&emsp; | &emsp;Full &emsp;| **70.5** / **81.3** | **48.3** / **73.5** | **50.1** / **58.0** |
> | &emsp;V2B-Tracker&emsp; | &emsp;Few&emsp; | 54.3 / 65.5 | 46.5 / 70.1 | 43.4 / 47.9 |
>
> It can be seen that without sufficient training data, the performance of the car and van categories is greatly reduced. However, the performance of the pedestrian category is slightly reduced. Since the motion of pedestrians is small between adjacent frames, using more training data might not improve the tracking performance significantly.
>
>
>
> &emsp;
>
> **Q3**: It will be helpful if authors can include videos.
>
> **A3**: We will provide the video link in the revised manuscript.
>
> &emsp;
>
> **Q4**: The meaning of different colors in Fig.3.
>
> **A4**: The different colors in Fig.3 indicate the features of different points. For example, in the template feature embedding module, for global feature, the rows with different colors mean the global feature vectors of different points.
>
> &emsp;
>
> **Q5**: The general guidance for parameter tuning on other new self-driving datasets.
>
> **A5**: For self-driving scenes, single object tracking methods mainly focus on tracking vehicles, cyclists and pedestrians. We can first pre-train the model with the parameters on KITTI or nuScenes and then fine-tune the parameters on the validation set of the new dataset with cross-validation.
>
> &emsp;
>
> **Q6**: Discuss the method can be extended to deal with RGB+LiDAR inputs.
>
> **A6**: In this paper, we focus on tracking 3D objects with point clouds. We would like to see how much performance can 3D object tracking methods achieve using only 3D point clouds. We believe that the tracking performance can be improved with the fusion of RGB+LiDAR inputs. For example, in our Siamese network for feature extraction, we can use a two-branch structure to extract point features and pixel features separately, and directly combine them to obtain fusion features for the subsequent target localization network. We will discuss it in the revised manuscript.

---

> > ### Comment · Reviewer_CfXd · 2021-08-28
> > **Discussion**
> >
> > I thank authors for providing well-structured rebuttals, and addressing most of the fellow reviewers' questions. My questions are mostly addressed as well. For Q2, I think this problem is worth discussing in future revisions of the paper, as I think this does sounds like a limitation of the proposed approach. For other questions, since authors promised addressing them in future revisions, I hope they keep the promise. Given those, I will retain my original rating, and happy to discuss more if needed.

---

### Official Review · Reviewer_m71r · 2021-07-17

**Rating:** 6
**Confidence:** 4

**Summary:**

This paper proposes a 3D object tracker that localizes the target object given its template and a small search space. Their proposed model largely consists of the following two modules: Siamese Shape-Aware Feature Learning Network and Voxel-to-BEV target localization. The feature learning network takes the template point clouds and the search space point clouds as input and generates a feature embedding for the search space, focusing on its correlation to the template point clouds and its ability to generate dense point clouds. The search space feature is then fed into Voxel-to-BEV target localization networks which flattens the 3D point clouds into BEV 2D image where the target localization happens. The proposed method outperforms the preceding works by a noticeable margin.

**Ethical Concerns:**

N / A

**Limitations And Societal Impact:**

I do not see a thorough analysis of the limitation or societal impact of this work. There is a limited study of failure cases in the supplementary material however as a tracking problem, I believe this approach has numerous limitations such as re-id and tracklet initialization. A couple sentences of a societal impact of this work seems to be expected by the conference as well.

**Main Review:**

Overall, while this paper outperforms the preceding works in terms of performance and computational cost, I have some concerns about the writing quality, novelty, and ablation study.

On the positive side, the proposed method clearly outperforms the preceding state-of-the-art methods. This paper has a proper baseline with improved performance, partial ablation study to support the usefulness of the proposed components, and thorough computational cost analysis for better real-world usefulness. I find it interesting that a point cloud generation task used only during the training phase helps the expressiveness of the point cloud features used for localization.

However, I have a few concerns about this paper:

First, the quality of the writing is poor. There are too many obvious grammatical errors and unpolished sentences. I believe this paper must be thoroughly proofread.

Second, the proposed Template Feature Embedding Networks is a huge complicated module with heavy engineering of features without detailed ablation study in itself. I do see a motivation behind the details of the network design however they are not properly backed by detailed ablation compared to its complexity. Especially considering the venue of the conference, I find it difficult to appreciate heavy engineering without proper theoretical study. For example, ablation of global / local features and different distance functions for correlation and similarity map may be useful.

Third, the Voxel-to-BEV target localization network seems to be heavily inspired by numerous preceding works in 3D object detection and I find it difficult to agree that this is a novel component. This paper must cite papers such as PointPillar and SECOND, which are commonly used 3D object detection encoder that look exactly the same as the proposed module.

**Time Spent Reviewing:**

3 hrs

---

> ### Author Response · Authors · 2021-08-12
> **Response to Reviewer m71r**
>
> We thank the reviewer for your valuable comments. Our responses to these comments are as below:
>
> **Q1**: Ablation study of template feature embedding.
>
> **A1**: We list the results of different settings as follows:
>
> | &emsp;&emsp;Settings | &emsp;&ensp;Car | Pedestrian | &emsp;&ensp;Van | &ensp;&nbsp;Cyclist |
> |:-:|:-:|:-:|:-:|:-:|
> | &emsp;local template&emsp; | 68.0 / 79.2 | 47.2 / 71.5 | 46.3 / 56.1 | 38.6 / 47.2 |
> | &emsp;global template&emsp; | 68.8 / 80.0 | 47.9 / 72.2 | 47.5 / 57.1 | 39.2 / 48.0 |
> | both | **70.5** / **81.3** | **48.3** / **73.5** | **50.1** / **58.0** | **40.8** / **49.7** |
>
> | &emsp;&emsp;Settings | &emsp;&ensp;Car | Pedestrian | &emsp;&ensp;Van | &ensp;&nbsp;Cyclist |
> |:-:|:-:|:-:|:-:|:-:|
> | Euclidean distance&emsp; | 70.1 / 80.9 | 49.1 / 73.7 | 49.5 / 57.3 | 40.6 / 49.5 |
> | cosine distance&emsp; | **70.5** / **81.3** | **48.3** / **73.5** | **50.1** / **58.0** | **40.8** / **49.7** |
>
> It can be seen that when simultaneously using local and global template embedding branches, our method can achieve the best results on all four categories. In addition, compared with the cosine distance, the results of using Euclidean distance are comparable to the results of using cosine distance.
>
> &emsp;
>
> **Q2**: The novelty of voxel-to-BEV target localization network compared with 3D detection methods such as PointPillar and SECOND.
>
> **A2**: 3D object detection needs to detect the bounding boxes of the objects in one frame while 3D single object tracking needs to detect the bounding box in the current frame with the given template. In the tracking sequence, since the 3D bounding box of the target in the template is given in advance, we can obtain the bounding box of the target of each frame by multiplying the coordinates of the bounding box of the target in the previous frame with the rotation matrix of the yaw angle. Therefore, we only need to regress the target center and yaw angle for subsequent frames in 3D single object tracking. 3D detection methods such as PointPillar and SECOND use the region proposal network with the BEV features to regress the 3D bounding boxes of the targets, which might fail due to the low-quality proposals on sparse point clouds. In order to avoid using the low-quality proposals on sparse point clouds for target center prediction, we directly regress the 3D center of the target with the highest response in the dense BEV feature map, where the dense BEV feature map is generated by voxelizing the learned dense geometric features and performing max-pooling along the z axis. Thus, with the constructed dense BEV feature map, for sparse point clouds, our method can more accurately localize the target center without any proposal. Actually, the SOTA tracking method P2B [12] utilizes the proposal based 3D object detection method VoteNet to vote the center in the sparse point clouds. The ablation study results in Table 5 of our paper demonstrate that our voxel-to-BEV location network can achieve better performance (**+5% gain**) than the VoteNet network in the **sparse scenarios** for single object tracking.
>
> &emsp;
>
> **Q3**: The approach has limitations such as re-id and tracklet initialization.
>
> **A3**: Our proposed method is a single object tracking method, where we directly regress the center of the target from the peak of the generated dense BEV map and update the template with the bounding box of the target in the previous frame. Therefore, we do not need to consider re-identification and tracklet initialization used in multiple object tracking.
>
> &emsp;
>
> **Q4**: Grammatical errors and unpolished sentences.
>
> **A4**: We will carefully correct the grammatical errors and polish the paper in the revised manuscript.

---

> > ### Comment · Reviewer_m71r · 2021-09-01
> > **Update**
> >
> > I appreciate authors for heavy efforts they made for the rebuttal. I appreciate the additional ablation study the authors provided.
> > I guess the biggest confusion came from "3D single object tracking" task that I am not familiar with. I feel like this problem is made up to avoid heavy competition of 3D object tracking, however I don't think I can make a strong argument against it. My concerns are partially relieved and I would like to update my rating accordingly.

---

### Official Review · Reviewer_KWCL · 2021-07-18

**Rating:** 4
**Confidence:** 3

**Summary:**

This paper suggests a framework named Siamese voxel-to-Bird Eye View (BEV) tracker, which aims to localise and follow each object of interest in sparse point clouds. The framework first feed the template and search area into the Siamese network to extract point features,  and then by adaptively learning the correlation between the template and search area, it embeds the template’s feature into the potential object in the search area . It also learns to encode shape-aware feature learning to characterize the shape information by generating a dense point cloud from a sparse point cloud. The suggested approach decomposes the 3D center into a 2D center and a z-axis center, and regress them in a dense bird’s eye view (BEV). The framework is claimed to achieve new state-of-the-art results on the KITTI tracking dataset.

**Limitations And Societal Impact:**

No, the authors didn't address limitations and societal impact.
Tracking 3D objects in point clouds has high potential impact in important applications, such as transportation  and self-driving vehicles. However, tracking technology can be generally deployed in human monitoring and surveillance as well which raise ethical and privacy issues.

**Main Review:**

1- Clarity: The paper is well written, and the idea is clear. The technical section can be followed easily. However, the application focus of the proposed technique is not very clear to me. The main claim in the paper is that it is a framework for tracking 3D object. is it single object or multiple object tracking? I don't see any component of tracking here such as track initiation, termination and occlusion handling. The framework seems to be a 3D detection method to me rather than any tracking frameworks. Many details clarifying how it is extended for tracking single/multiple objects are not explained.   Considering the dataset used for the evaluation, eg KITTI, I am not convinced that how it is used for tracking single object only and how it is compared with the other methods in the KITTI leaderboards.

2- Many missing related works: The related works and the comparing frameworks seem to be very narrow and cherry picked. There exist many related works in 2D MOT community (e.g. MOTChllenge or KITTI) are missed here. Similarly for 3D MOT, there exist many state-of-the work methods in KITTI, NuScene, Waymo and JRDB datastes and the proposed framework could be compared with. To name just few:
X. Weng, J. Wang, D. Held and K. Kitani. 3D Multi-Object Tracking: A Baseline and New Evaluation Metrics. IROS 2020 (KITTI, JRDB, NuScene ).
A. Shenoi, M. Patel, J. Gwak, P. Goebel, A. Sadeghian, H. Rezatofighi, R. Martin-Martin and S. Savarese: JRMOT: A Real-Time 3D Multi-Object Tracker and a New Large-Scale Dataset. IROS 2020 (KITTI, JRDB ).
A. Kim, A. Ošep, L. Leal-Taixé. EagerMOT: 3D Multi-Object Tracking via Sensor Fusion. ICRA 2021(KITTI,  NuScene ).


3 -Contribution: the paper does not offer any new theoretical contribution. It is rather a pure application paper using built upon the existing  machine learning tools with some tweaks and minor extensions. While the application papers are also well acknowledged by the NeurIPS research community, but such papers require comprehensive experimental evaluation on different benchmark datasets and comapring with many state-of-the-art frameworks to validate the efficacy of their proposed system. Without this, I would be hesitant to vote for the acceptance of such a paper in this venue.

4- Experiments and datasets: The experiments are tested on KITTI dataset  only using a new metric (not those used by this benchmark for MOT). Why the approach was not evaluated on KITTI test set and using KITTI tracking metrics such as MOTA or HOTA? I it is not clear to me how the framework can deal with non-rigid object like human (how template is defined) and if it can track human as non-rigid object why it is not tested on 3D human tracking datasets such as JRDB. What about other 3D tracking self-driving datasets such as NuScene or Waymo?



**Time Spent Reviewing:**

2

---

> ### Author Response · Authors · 2021-08-10
> **Response to Reviewer KWCL**
>
> We thank the reviewer for taking the time to review our paper. Our responses to the comments on multiple object tracking are as below:
>
> **Q1**: Is it 3D single object or 3D multiple object tracking? How is it used for tracking single object?
>
> **A1**: Our method is a tracking-by-detection method for 3D single object tracking. Our method consists of the Siamese network for feature extraction and localization network for center and yaw angle prediction. The Siamese network aims to learn dense geometric features of the potential target with sparse points in the search area by measuring the similarity between the template and search area in the embedding space. Since the 3D bounding box (center, size, and yaw angle) of the target in the template is given in advance, the localization network only needs to regress the target center and yaw angle. Once the target center and yaw angle of the target are predicted, we can determine the bounding box of the target. When starting single object tracking, given the template in the first frame, we first generate the search area in the second frame according to the target position in the first frame, and then feed the template and search area into the network to localize the potential target in the search area. After that, we update the template by combining the template with the points located in the bounding box of the localized target. For the subsequent frames, we alternately perform this process between two adjacent frames to track the target frame by frame. Therefore, as a single 3D object tracking method, we do not need to consider re-identification and termination in multiple object tracking.
>
> &ensp;
>
> **Q2**: How is it compared with the other methods in the KITTI leaderboards?
>
> **A2**: The KITTI leaderboards are used for evaluating multiple object tracking (MOT) methods. Since our method is a single object tracking method, we only compare our method with SOTA single object tracking methods, such as SC3D [13] and P2B [12]. Following SC3D and P2B, we use the KITTI training set (21 video sequences) for 3D single object tracking by generating a tracklet for each instance in the scenes, where scenes 0-16 are used for training, scenes 17-18 for validation, and scenes 19-20 for testing.
>
> &ensp;
>
> **Q3**: Missing related works of multiple object tracking.
>
> **A3**: In the paper, we listed related works of single object tracking and will add related works of multiple object tracking in Sec. 2 in the revised manuscript.
>
> &ensp;
>
> **Q4**: Contribution of the paper.
>
> **A4**: The main contribution of our proposed method is two-fold. On the one hand, in order to track the target with sparse points, we propose the target completion model (shape-aware feature learning module) to learn dense geometric features of the potential target from the samples of search areas, where the global-local branch is constructed with the gate mechanism to enhance the feature of the potential target and suppress the background in the search area. Compared with the simple template completion model in SC3D [13], our method can better recover complex geometric structures of the potential target. On the other hand, in order to avoid using the low-quality proposals on sparse point clouds for target center prediction, we develop a simple yet effective target center localization model without any proposal. The center of the potential target can be directly regressed from the dense BEV feature map, which is generated by voxelizing the learned dense geometric features and performing max-pooling along the z axis. Compared with P2B [12], which utilizes proposal based VoteNet to vote the center in the sparse point clouds, our method can more accurately regress the target center from the dense BEV feature map than the sparse point clouds. Experimental results on the KITTI tracking dataset demonstrate that our method can achieve **SOTA performance** (with **+7% gain**) and can handle object tracking in **sparse scenes**.
>
> &ensp;
>
> **Q5**: Use KITTI leaderboards’ tracking metrics such as MOTA or HOTA.
>
> **A5**: MOTA and HOTA are usually used in multiple object tracking evaluation and not used in single object tracking evaluation. For a fair comparison, following single object tracking methods SC3D [13] and P2B [12], we also adopt the Success and Precision metrics to evaluate our method, where Success indicates the IoU between the predicted and ground truth bounding boxes while Precision indicates the error AUC of the distance between the centers of two bounding boxes.
>
> &ensp;
>
>
> **Q6**: How the framework can deal with non-rigid object like human and if it can track human as non-rigid object why it is not tested on 3D human tracking datasets such as JRDB?
>
> **A6**: In our 3D single object tracking method, once the center and yaw angle of the target in the search area are predicted, we can obtain the bounding box of the target by multiplying the coordinates of the bounding box of the target in the previous frame with the rotation matrix of the yaw angle. Therefore, our method can mainly handle rigid object tracking. Actually, since in the self-driving scenes the non-rigid motion of pedestrian is usually small, our rigid object tracking method can still effectively track pedestrians. Since JRDB is a multiple object tracking dataset, we do not conduct human tracking experiments on it.
>
> &ensp;
>
> **Q7**: What about other 3D tracking self-driving datasets such as nuScenes?
>
> **A7**: To demonstrate the effectiveness of our method on other 3D self-driving datasets, we conduct the experiment on the nuScenes dataset. Since the ground truth of the online test set of nuScenes for multiple object tracking cannot be obtained, we adopt the nuScenes training set (700 sequences) and validation set (150 sequences) for 3D single object tracking by generating a tracklet for each instance in the scenes. In the experiment, 700 sequences of the training set are used for training, and 150 sequences of the validation are used for testing. The results on the car and pedestrian categories are shown as follows:
>
>
> | &emsp;&emsp;&ensp;Method&ensp;&ensp;&ensp; |  &ensp;&ensp;&ensp;Car | Pedestrian |
> |:-:|:-:|:-:|
> |  SC3D[13]       |   37.3 / 43.8   | 13.6 / 33.2 |
> |  P2B[12]        |   40.2 / 47.0   | 21.3 / 47.8 |
> |  V2B-Tracker (ours)  |   **46.0** / **54.8**   | **27.9** / **53.6** |
>
>
> It can be seen that our method can achieve better performance than the other two methods on the nuScenes dataset. We will add the results of the nuScenes dataset in the revised manuscript.

---

### Official Review · Reviewer_1Mqw · 2021-07-18

**Rating:** 7
**Confidence:** 4

**Summary:**

This paper proposes a novel Siamese voxel-to-BEV tracker for object tracking in sparse 3D point clouds. The main contributions are two folds: It introduces a shape-aware feature learning network to enhance the discrimination of targets in the search area; It develops a voxel-to-BEV target localization network to better localize 3D objects. The proposed method achieves the state-of-the-art performance compared with previous arts and ablation studies demonstrate the efficacy of various designs.

**Limitations And Societal Impact:**

As I mentioned previously, some of the claims are not fully justified (better at handling sparse points, necessary to use both local and global features in the template feature embedding module and in the shape-aware feature learning module). Also the ablation study only considers the car category, which is not really representative enough. All four categories should be considered to really making the claimed contributions convincing enough. I hope authors could address this issue during rebuttal. Some claims sound awkward to me, such as line 378-380 (stable does not mean accurate.) The paper should point out that this proposed method is not suitable for indoor scenes where a BEV map might not be sufficient for object proposal.

**Main Review:**

The submission is addressing an important problem, namely object tracking in 3D point clouds. The submission is clearly written, making it easy to follow. The content is also well organized with enough technical details provided. Generally speaking, the submission is technical sound. The motivation behind various designs is clear and the experimental results partially support some of the claims.

However, there are also claims not fully justified experimentally. For example, the submission claims it is better at handling sparse point clouds compared with previous methods including SC3D and P2B. If we compare Table 1 and Table 2, when switching from all types of objects to sparse objects, only the proposed method suffers from a performance drop while all the other two baseline methods actually get better performance. Therefore, it is not clear whether the claimed shape aware feature really makes the proposed method more suitable for sparse point clouds. Another example is that there's no ablation studies justifying the necessity of using both global and local features in the template feature embedding module and in the shape-aware feature learning module.

The method is quite new and its difference from previous works are also clearly discussed. The related work is adequate as far as I can tell. The overall experimental results are quite impressive, which outperforms previous methods by a large margin.

**Time Spent Reviewing:**

3

---

> ### Author Response · Authors · 2021-08-10
> **Response to Reviewer 1Mqw**
>
> We thank you for acknowledging that our method is quite new and different from previous works. We also thank you for your valuable comments to improve our experiment. The experimental results can be found as below:
>
> **Q1**: When switching from all types of frames to sparse frames, the proposed method suffers from a performance drop while the other two baseline methods actually get better performance. It is not clear whether the claimed shape aware feature really makes the proposed method more suitable for sparse point clouds.
>
> **A1**: Since in the single object tracking method the template in the current frame will be updated with the localization of the target in the previous frame, the inaccurately updated template will lead to poor tracking performance on the consecutive frames. According to Tables 1 and 2 in the main paper and Table 1 in the supplementary material, the average tracking results (Success/Precision) of the four categories (car, pedestrian, van, and cyclist) on the sparse frames in the KITTI dataset are 31.2 / 48.5 (SC3D), 42.4 / 60.0 (P2B) and **58.4** / **75.2** (our V2B-Tracker). Our V2B-Tracker outperforms SC3D and P2B with a large margin (over 15%) on the sparse frames. The poor tracking performance of SC3D and P2B on large amounts of sparse frames (the proportion of sparse frames in all test frames is up to 68%) leads to the inaccurate template updates on the consecutive dense frames. Thus, SC3D and P2B cannot obtain better tracking performance on the dense frames. Although SC3D uses template shape completion, due to limited template samples and large variations of the potential target in the search area, it cannot accurately recover the complex geometric structures of the target in the sparse frames, which poses challenges on localizing the potential target with sparse points. On the contrary, our V2B-Tracker employs the proposed shape-aware feature learning module to generate dense and complete point clouds of the potential target for the target shape completion, leading to more accurate localization of the target in the sparse frames. Thus, our method performs better tracking results in the dense frames for the KITTI dataset.
>
> &ensp;
>
> **Q2**: Ablation study of both local and global features in the template feature embedding module and shape-aware feature learning module on four categories.
>
> **A2**: We list the ablation study results of different settings as follows:
>
>
> | &emsp;&emsp;&emsp;&emsp;&emsp;&emsp;Settings&emsp;&emsp;| &emsp;&ensp;Car&emsp; | &emsp;Pedestrian&emsp; | &emsp;&ensp;Van &emsp;| &emsp;&ensp;Cyclist&emsp; |
> |:-:|:-:|:-:|:-:|:-:|
> | &emsp;&emsp;&emsp;&emsp;&emsp;local template&emsp;&emsp;&emsp;&emsp;&ensp; | 68.0 / 79.2 | 47.2 / 71.5 | 46.3 / 56.1 | 38.6 / 47.2 |
> | &emsp;&emsp;global template&emsp;&emsp; | 68.8 / 80.0 | 47.9 / 72.2 | 47.5 / 57.1 | 39.2 / 48.0 |
> | both | **70.5** / **81.3** | **48.3** / **73.5** | **50.1** / **58.0** | **40.8** / **49.7** |
>
>
> | &emsp;&emsp;&emsp;&emsp;&emsp;&emsp;Settings &emsp;&emsp;| &emsp;&ensp;Car&emsp; | &emsp;Pedestrian&emsp; | &emsp;&ensp;Van &emsp;| &emsp;&ensp;Cyclist&emsp; |
> |:-:|:-:|:-:|:-:|:-:|
> | &emsp;&emsp;&nbsp;local geometric of target&emsp;&emsp;&nbsp; | 68.6 / 79.3 | 47.8 / 72.0 | 48.7 / 56.4 | 39.7 / 48.4 |
> | &emsp;&emsp;global shape of target&emsp;&emsp; | 69.6 / 80.3 | 47.9 / 71.0 | 49.2 / 57.5 | 40.5 / 48.9 |
> | both | **70.5** / **81.3** | **48.3** / **73.5** | **50.1** / **58.0** | **40.8** / **49.7** |
>
> It can be clearly seen that when we simultaneously employ local and global features in the two modules, our method can achieve the best performance on four categories.
>
> &ensp;
>
> **Q3**: Provide results of ablation study in all four categories in Table 4 of the main paper.
>
> **A3**: We list the ablation study results of four categories as follows:
>
> | &emsp;&emsp;&emsp;&emsp;&emsp;V2B-Tracker&emsp;&emsp;&emsp;| &emsp;&ensp;Car&emsp; | &emsp;Pedestrian&emsp; | &emsp;&ensp;Van &emsp;| &emsp;&ensp;Cyclist&emsp; |
> |:-:|:-:|:-:|:-:|:-:|
> | w/o template feature embedding | 63.9 / 73.9 | 42.6 / 65.6 | 42.3 / 49.1 | 33.2 / 42.4 |
> | w/o shape-aware feature learning | 67.6 / 78.2 | 44.0 / 69.0 | 44.2 / 51.9 | 36.5 / 45.9 |
> | default| **70.5** / **81.3** | **48.3** / **73.5** | **50.1** / **58.0** | **40.8** / **49.7** |
>
> It can be found that without using template feature embedding or shape-aware feature learning, the performance of our V2B-Tracker is greatly reduced.
>
> &emsp;
>
> **Q4**: The paper should point out that this proposed method is not suitable for indoor scenes where a BEV map might not be sufficient for object proposal.
>
> **A4**: Due to the complex geometry relationship between multiple objects in indoor scenes, the BEV feature map might not capture complex context information well to accurately localize the object. We will discuss it in the revised manuscript.

---

> > ### Comment · Reviewer_1Mqw · 2021-08-23
> > **Reviewer's Response**
> >
> > Thanks for addressing my concerns and I would like to keep my original rating.

---

### Official Review · Reviewer_QKmE · 2021-07-19

**Rating:** 6
**Confidence:** 4

**Summary:**

This paper tackles the problem of single-object tracking from point clouds. Specifically, it adopts a tracking by detection framework to find the target in the search area, implemented with a siamese network that extracts both the template and search area features. To improve feature discriminativeness, the template feature is embeded to the search area feature, trained with a regularization shape completion loss. In terms of target localization, in contrast to previous proposal based method, this paper adopts a dense detection approach that directly localizes the target from dense BEV voxel locations. Experiments on KITTI dataset with custom split shows significant improvements over previous state-of-the-art methods.

**Limitations And Societal Impact:**

I didn't see specific discussions about the limitations and potential negative societal impact in the submission, please add these in the revised submission.

**Main Review:**

Following aspects of the submission can be improved.

1. Needs a clear comparison with previous methods.
The method proposed in the submission is largely based on two previous methods, P2B [12] and SC3D [13]. In particular, the overall two-step framework follows P2B, where the first step embeds the template feature into the search area feature, implemented by a Siamese network; and the second step localizes the target given the enhanced search area feature. The main difference lies in the second step, where P2B uses a proposal based method, and this paper adopts a dense detection method, which is shown by experiments to perform much better on objects with sparse points. A suggestion here would be to add qualitative results in addition to the Table 5 in the submission to show examples of these improvements.
Another claimed novelty of shape-awre feature learning also largely follows the scene completion regularization loss proposed in SC3D, with the difference being the specific input feature (Line 187-190). A suggestion here would be to add a specific ablation study for the changes made on top of the regularization loss from SC3D, so as to justify these modifications and thereof the technical novelty.
Overall, I would suggest authors provide a clear comparison with previous methods that are technically related, and clarify what's inherited from them, and what's changed to better solve an existing problem.

2. Needs better presentation of the method section.
It's hard to follow the method section as tons of technical details are thrown to the readers without providing enough context and motivation of why the architecture is designed so. I would strongly suggest the authors to re-write the method section in the revised submission.

3. More results in the experiments.
There can be more results in the experiments to help readers understand how the model is working. For example, the ablation study with SC3D as mentioned in point 1, qualitative comparison with both SC3D and P2B in all four categories, especially the cyclist because the proposed method performs much worse in this category compared with SC3D. Also, in Table 3 the speed comparison, the official speed of P2B is 40FPS on a NVIDIA 1080Ti GPU, however, Table 3 reports 17FPS on a NVIDIA TITAN RTX GPU, any reason why this huge gap?

**Time Spent Reviewing:**

2

---

> ### Author Response · Authors · 2021-08-10
> **Response to Reviewer QKmE**
>
> We thank the reviewer for the valuable comments. The detailed responses to the comments can be found as below:
>
> **Q1**: A clear technical comparison with P2B[12] and SC3D[13].
>
> **A1**: Although Our method and P2B[12] are tracking-by-detection methods, the main differences between them are two-fold, including the Siamese network for feature extraction and localization network for center prediction. In the Siamese network for feature extraction, P2B employs the global template feature embedding to extract the features of the sparse point clouds of the potential target. Nonetheless, our method first employs the global and local template feature embedding to extract the sparse features of the potential target and then introduces a shape aware feature learning module to learn the dense geometric features of the potential target, where the complete and dense point clouds of the target are generated. Thus, the geometric structures of the potential target can be captured better so that the potential target can be effectively distinguished from the background in the search area. In the localization network for center prediction, P2B uses the proposal based VoteNet to vote the center of the potential target, which might fail due to low-quality proposals on sparse point clouds. Our method adopts a dense detection method for center localization without any proposal, where the dense BEV feature map generated from the learned dense geometric features is utilized to more accurately regress the center of the potential target. We list the ablation study of the two components as follows:
>
> | &emsp;&emsp;&emsp;&emsp;&emsp;V2B-Tracker &emsp;&emsp;| &emsp;&ensp;Car&emsp; | &emsp;Pedestrian&emsp; | &emsp;&ensp;Van &emsp;| &emsp;&ensp;Cyclist&emsp; |
> |:-:|:-:|:-:|:-:|:-:|
> | Siamese network in P2B&emsp; | 67.3 / 78.0 | 46.9 / 70.9 | 46.4 / 55.5 | 37.2 / 46.0 |
> | our Siamese shape-aware network&emsp; | **70.5** / **81.3** | **48.3** / **73.5** | **50.1** / **58.0** | **40.8** / **49.7** |
>
> | &emsp;&emsp;&emsp;&emsp;&emsp;V2B-Tracker &emsp;&emsp;| &emsp;&ensp;Car&emsp; | &emsp;Pedestrian&emsp; | &emsp;&ensp;Van &emsp;| &emsp;&ensp;Cyclist&emsp; |
> |:-:|:-:|:-:|:-:|:-:|
> | &emsp;&emsp;&emsp;&emsp;VoteNet in P2B&emsp;&emsp;&emsp;&emsp;&emsp;&ensp;  | 62.7 / 75.9 | 32.8 / 62.0 | 34.4 / 41.9 | 34.3 / 41.5 |
> | &emsp;&emsp;&emsp;&emsp;our Voxel-to-BEV&emsp;&emsp;&emsp;&emsp;&emsp; &ensp; | **70.5** / **81.3** | **48.3** / **73.5** | **50.1** / **58.0** | **40.8** / **49.7** |
>
> It can be seen that our proposed Siamese network and voxel-to-BEV target center localization network are superior to P2B on all four categories, where the Siamese network and VoteNet in P2B are used in our framework, respectively.
>
> SC3D [13] is a template matching method for object tracking by comparing the similarity of the latent vectors between the template and candidate bounding box proposal, while our method is a tracking-by-detection method for object tracking by directly regressing the target center in the search area from the learned geometric features of the potential target. Although shape completion is used in both methods, the main differences between them are also two-fold. On the one hand, SC3D uses a simple auto-encoder structure to learn the template completion model from the template samples (given bounding boxes of the targets), while our method is a complex point cloud generation method that learns the target completion model (shape-aware feature learning module) from the samples of search areas with the gate mechanism to enhance the feature of the potential target and suppress the background in the search area. Due to limited templates and large variations of the potential target in the search area, the template completion model in SC3D cannot recover complex geometric structures of the potential target well, which might lead to the inaccurate matching. On the other hand, the template completion model in SC3D only employs PointNet to extract point features of sparse point clouds, while our target completion model constructs a global-local branch to extract global shape features and local geometric features of sparse point clouds. We list the ablation study of shape completion as follows:
>
> | &emsp;&emsp;&emsp;&emsp;&emsp;V2B-Tracker &emsp;&emsp;| &emsp;&ensp;Car&emsp; | &emsp;Pedestrian&emsp; | &emsp;&ensp;Van &emsp;| &emsp;&ensp;Cyclist&emsp; |
> |:-:|:-:|:-:|:-:|:-:|
> | shape completion in SC3D&emsp;&emsp; | 67.3 / 78.4 | 46.1 / 70.2 | 47.5 / 56.6 | 36.9 / 45.4 |
> | our shape-aware feature learning&emsp;&emsp; | **70.5** / **81.3** | **48.3** / **73.5** | **50.1** / **58.0** | **40.8** / **49.7** |
>
> It can be seen that our shape-aware feature learning model outperforms the completion model in SC3D on all four categories, where template shape completion in SC3D is used in our framework.
>
> &ensp;
>
>
> **Q2**: Provide enough context and motivation for architecture design in the method section.
>
> **A2**: Our single object tracking method mainly consists of two components, i.e., Siamese shape-aware feature learning network and voxel-to-BEV localization network. The Siamese network aims to learn dense geometric features of the potential target with sparse points in the search area. By voxelizing the learned geometric features to form the dense BEV feature map, we can employ the localization network to directly regress the center and yaw angle of the target for tracking. Specifically, in the Siamese network, we first employ template feature embedding to encode the search area by learning the similarity of the global shape and local geometric structures between the template and search area. Due to the sparse and incomplete point clouds of the potential target in the search area, we then employ shape-aware feature learning to learn dense geometric features of the target, where the dense and complete point clouds of the target can be obtained. It is expected that the learned features from the generated dense point clouds can characterize the geometric structures of the target better. In the localization network, in order to improve the localization precision in sparse point clouds, we utilize the voxelization and max-pooling operation to convert the discriminative features of sparse 3D points into the dense BEV feature map for the target localization. We will provide detailed context and motivation for architecture design in the revised manuscript.
>
> &ensp;
>
> **Q3**: The huge gap between the official speed in P2B[12] (45.5FPS, 1080Ti GPU) and the reported speed in Table 3 (17FPS, TITAN RTX GPU).
>
> **A3**: As stated in [12], the official running time of P2B consists of 7.0ms (data loading), 14.3ms (network inference), and 0.9ms (post-processing) per frame. In this paper, the reported running time consists of 43.1ms (data loading), 9.53ms (network inference), and 6.2ms (post-processing) per frame. Since the TITAN RTX GPU is better than 1080Ti, the network inference speed in our paper is faster than the official speed of P2B. However, since our server cluster has large communication delays and uses the virtual cutting technique to divide multi-core CPUs, the reported data loading time and post-processing time in our paper is slower than the official running time in P2B. Therefore, in terms of FPS, there is a gap between our reported speed in our paper and the official speed in P2B.

---

> > ### Comment · Reviewer_QKmE · 2021-08-29
> > **Reviewer's final decision**
> >
> > Thanks to the authors for addressing my comments, and I decide to keep my original rating.

---

### Decision · Program_Chairs · 2021-09-27

**Decision:**

Accept (Poster)

**Comment:**

Initially, one of the reviewers expressed concerns about the paper (lack of clarity and limited novelty) and ranked the paper below acceptance. Another area of concern was related to a number of claims that the authors made and were not fully justified experimentally. As the ensuing rebuttal managed to successfully address most of reviewer’s concerns, ACs and the majority of the reviewers agreed that this is a strong paper that deserves acceptance.   Authors are highly encouraged to address the key comments reported by reviewers as well as to implement all the improvements (as indicated by authors in the rebuttal, including addressing the over statements/claims as discussed above) in the final camera-ready version.